# On a Cold Night: Transcriptomics of Grapevine Flower Unveils Signal Transduction and Impacted Metabolism

**DOI:** 10.3390/ijms20051130

**Published:** 2019-03-05

**Authors:** Mélodie Sawicki, Marine Rondeau, Barbara Courteaux, Fanja Rabenoelina, Gea Guerriero, Eric Gomès, Ludivine Soubigou-Taconnat, Sandrine Balzergue, Christophe Clément, Essaïd Ait Barka, Nathalie Vaillant-Gaveau, Cédric Jacquard

**Affiliations:** 1Unité de Recherche Résistance Induite et Bioprotection des Plantes—EA 4707, Université de Reims Champagne-Ardenne, UFR Sciences Exactes et Naturelles, SFR Condorcet FR CNRS 3417, Moulin de la Housse—Bâtiment 18, BP 1039, 51687 REIMS Cedex 2, France; me.sawicki@laposte.net (M.S.); marine.rondeau@univ-reims.fr (M.R.); barbara.courteaux@univ-reims.fr (B.C.); clarisse.rabenoelina@univ-reims.fr (F.R.); christophe.clement@univ-reims.fr (C.C.); ea.barka@univ-reims.fr (E.A.B.); nathalie.vaillant-gaveau@univ-reims.fr (N.V.-G.); 2Luxembourg Institute of Science and Technology (LIST), Environmental Research and Innovation (ERIN) Department, 41 rue du Brill, L- 4422 Belvaux, Luxembourg; gea.guerriero@list.lu; 3Institute of Vine and Wine Sciences, UMR 1287 Ecophysiology and Grape Functional Genomics, University of Bordeaux, INRA 210 Chemin de Leysotte - CS 50008, 33882 Villenave d’Ornon CEDEX, France; eric.gomes@inra.fr; 4Institute of Plant Sciences Paris Saclay IPS2, CNRS, INRA, Université Paris-Sud, Université Evry, Université Paris-Saclay, Bâtiment 630, 91405 Orsay, France; ludivine.soubigou-taconnat@inra.fr (L.S.-T.); sandrine.balzergue@inra.fr (S.B.); 5Institute of Plant Sciences Paris-Saclay IPS2, Paris Diderot, Sorbonne Paris-Cité, Bâtiment 630, 91405 Orsay, France; 6IRHS, INRA, AGROCAMPUS-Ouest, Université d’Angers, SFR 4207 QUASAV, 42 rue Georges Morel, 49071 Beaucouzé CEDEX, France

**Keywords:** *Vitis vinifera*, flower, candidate genes expression, cold stress, signaling cascades, secondary metabolism, cell wall

## Abstract

Low temperature is a critical environmental factor limiting plant productivity, especially in northern vineyards. To clarify the impact of this stress on grapevine flower, we used the *Vitis* array based on Roche-NimbleGen technology to investigate the gene expression of flowers submitted to a cold night. Our objectives were to identify modifications in the transcript levels after stress and during recovery. Consequently, our results confirmed some mechanisms known in grapes or other plants in response to cold stress, notably, (1) the pivotal role of calcium/calmodulin-mediated signaling; (2) the over-expression of sugar transporters and some genes involved in plant defense (especially in carbon metabolism), and (3) the down-regulation of genes encoding galactinol synthase (GOLS), pectate lyases, or polygalacturonases. We also identified some mechanisms not yet known to be involved in the response to cold stress, i.e., (1) the up-regulation of genes encoding G-type lectin S-receptor-like serine threonine-protein kinase, pathogen recognition receptor (*PRR5*), or heat-shock factors among others; (2) the down-regulation of Myeloblastosis (MYB)-related transcription factors and the Constans-like zinc finger family; and (3) the down-regulation of some genes encoding Pathogen-Related (PR)-proteins. Taken together, our results revealed interesting features and potentially valuable traits associated with stress responses in the grapevine flower. From a long-term perspective, our study provides useful starting points for future investigation.

## 1. Introduction

Agricultural systems are vulnerable to climatic variability. To cope with adverse conditions, plants have evolved a range of responses through multifaceted signaling pathways at molecular, physiological, and biochemical levels. Because of this, plants are able to survive and adapt to unfavorable conditions [1]. Consequently, different groups of stress-inducible genes and gene products are involved in complex signal transduction networks. A first group contains regulatory proteins involved in further regulation of signal transduction and gene expression, such as transcription factors (TFs), protein kinases, protein phosphatases, different enzymes, and other signaling molecules, e.g., calmodulin-binding protein. A second group includes proteins involved in the stress tolerance response, such as late-embryogenesis abundant (LEA) proteins, heat shock proteins (HSPs), osmoprotectant biosynthesis-related proteins, carbohydrate metabolism-related proteins, water-channel proteins, sugar/potassium transporters, detoxification enzymes, proteases, senescence-related genes, as well as protease inhibitors [2].

Among various environmental stresses, low temperature is one of the most critical factors limiting the productivity and geographical distribution of plants. Low temperatures lead to decreased yields, stunted growth, and tissue damages during crop production [3] and can severely hamper reproductive development [4]. A generic pathway for the plant response to cold stress has been described in the literature. Briefly, an extracellular cold stress signal is initially perceived by membrane receptors/sensors, which in turn activate large and complex intracellular signaling networks leading to cold stress tolerance, either directly or indirectly. Indeed, plants have various ways to respond to cold stress, including changes in the composition, structure, and function of the plasma membrane and cell wall, cryoprotectant molecules’ synthesis, as well as an increase in the reactive oxygen species’ (ROS) scavenging activity [5,6,7,8]. Cold temperatures also lead to modifications of primary and secondary metabolic pathways [9,10,11]. Low temperatures activate a number of cold-inducible genes, such as those encoding dehydrins, lipid transfer proteins, translation elongation factors, and LEA proteins [4], as well as molecular chaperones’ HSPs [12,13]. LEA and HSPs are known to be involved in protecting macromolecules, such as enzymes and lipids [14]. In response to cold stress, several mechanisms are involved, including calcium-mediated signal transduction [15,16]. This results in cold-specific pathway activation and the induction of *ICE* (inducer of CBF Expression 1) expression and subsequently of CBF (C-repeat/DRE-Binding Factor) transcription factors, as well as Abscisic acid (ABA)-related metabolism activation [17]. The response to cold stress also includes an accumulation of osmoprotectants, such as sugars [18], amino acids (namely proline [19]), or also post-transcriptional regulation of proteins [20]. Abscisic acid (ABA) may also function as a secondary signal [15] and several TFs also respond to cold signals at the early stage during low-temperature exposure [21].

The grapevine is one of the most widely cultivated fruit crops around the world and its culture has a strong economic importance in many countries. During the growing season, the grapevine is relatively sensitive to cold [22]. Although the flower, which forms the fruit, is a determinant factor for the yield, relatively few studies are devoted to this organ. During the reproductive phase, meiosis is a particularly sensitive key stage. In the grapevine flower, female meiosis coincides with drastic physiological changes in the whole plant. At this time, carbon nutrition switches from the mobilization of reserves from perennial organs to photosynthesis in the leaves [23]. During this delicate period, any interruption or partial decline of the sugar supply, as a result of environmental stress, might lead to extensive flower abortion [24,25,26].

We recently showed that the grapevine flower is able to cope with cold temperatures by adapting its carbohydrate metabolism, using mechanisms differentially induced according to the stress intensity [27]. We also showed that the contents of minor sugars are altered when grapevine plants are exposed to cold. It has been shown that trehalose and trehalose-6-phosphate contents fluctuate in in vitro plantlets [28], but also in flowers exposed to a cold night at female meiosis [29], making these sugars putative signaling molecules implied in the cold stress response. Although leaves are the principal source of photosynthates, the reproductive structures of many plant species are also reported to be photosynthetically active, assimilating significant amounts of carbon [30] to partly compensate for reproduction costs. In the grapevine, we demonstrated that the inflorescence shows photosynthetic activity [31] and is able to assimilate and export the majority of the assimilated carbon, thus playing a crucial role in carbon balance by sustaining the early development of leaves [32].

In temperate vineyards, cold nights can occur in late spring at the time of female meiosis in grapevine flowers. According to Tattersall et al. [33], chilling stress induces larger and more complex responses of the grapevine leaf transcriptome than any other abiotic stresses. Previous studies reported that transcripts related to metabolism, transport, signal transduction, and transcription were more abundant in grapevine leaves in response to cold stress [34], and that the ability to actively respond to cold stress contributes to the higher tolerance in cold-tolerant grapevines [35]. Nevertheless, there is no available information on the global changes in the gene expression in grapevine flowers exposed to cold stress. This step is crucial to gaining a better understanding of the flower abortion mechanisms mainly related to cold stress in perennial crops. To fill such a gap and to identify the signaling pathways involved in the cold response, we used a genome-wide grapevine microarray approach to investigate the gene expression responses of flowers submitted to a cold night. Such an investigation of the flower molecular response should finally suggest candidate genes for a genetic improvement of the cold tolerance in agronomic crops. Moreover, with the exception of a previous study addressing cold-induced transcriptional changes in the buds of the blueberry [36], the molecular responses to cold stress are still poorly characterized at the flower level in plants. In particular, to our knowledge, the present study is the first to explore the effects of cold on the flowers of the grapevine. Consequently, our objectives were to identify modifications at the transcript level (i) after a cold night at the female meiosis time and (ii) during recovery (2 h after the end of the cold night). Our results corroborated some mechanisms known in the plant’s response to cold stress, but also identified the involvement of some mechanisms so far not reported to respond to cold stress. These results give a global overview of the main functional gene categories impacted by cold-stress, and finally provide a useful starting point for future investigations of stress exposure in early flower development.

## 2. Results and Discussion

### 2.1. Expression Analysis and Validation of the Data Set

All transcripts significantly differentially expressed at least in one time-point between the control and cold stress were extracted, yielding a total of 1304 differentially expressed genes (DEGs), including 169 in common and divided into eight different groups (Figure 1). The Venn diagram shows the number of transcripts modulated by cold at 8 h and 2 h post-stress (PS) (Figure 1). Among the genes showing differential expression at 8 h, 153 were up-regulated, 101 were down-regulated, while 528 were induced and 353 were repressed 2 h PS, compared to their corresponding controls, respectively. A total of 63 transcripts were commonly up-regulated at 8 h and 2 h PS, whereas 90 were commonly repressed. Moreover, 6 transcripts were up-regulated at 8 h and down-regulated at 2 h PS, while 10 were down-regulated at 8 h and up-regulated at 2 h PS.

To validate the results obtained with the microarray analysis, we carried out quantitative real-time RT-PCR (qRT-PCR) assays on 15 genes (the primers can be found in Appendix A) randomly chosen in the 8 transcriptomic categories. Results on these 15 selected genes confirmed the changes in transcript abundance (Figure 2). Moreover, the coefficient obtained by plotting the microarray vs. RT-qPCR (*R*^2^ = 0.87) indicated a good overall agreement between the two methods, thereby confirming the validity of the microarray results (Appendix A).

### 2.2. Functional Analyses of the Transcriptome Changes in Response to a Cold Night

The major functional categories highly impacted by the cold night at 8 h PS are reported in Table 1.

### 2.3. Cold Stress Perception by the Grapevine Flower and Signal Transduction

During the last years, cold acclimation, an important process in plant survival, has been particularly studied [37]. Although different, some mechanisms involved in this process are common to modifications involved by cold stress [10,18,38,39,40]. The identity of specific temperature sensors is not known, but multiple primary sensors may be involved [10,41]. Then, the signaling network is activated in the cell, first with the generation of secondary signaling molecules, such as calcium (Ca^2+^) and calmodulin (CaM), as well as ROS. These secondary molecules can initiate protein phosphorylation cascades, such as (Ca^2+^)-dependent protein kinases (CDPK) and mitogen-activated protein kinases (MAPK) cascades, which activate numerous TFs and specific stress-responsive genes [4,18,42,43,44]. All of these messengers promote the production of cold-responsive proteins, such as (i) regulatory proteins controlling the cold stress signal transduction, e.g., the CBF/DREB (C-repeat binding factor/dehydration-responsive element binding) responsive pathway [45,46,47]; (ii) proteins functionally involved in the resistance/tolerance response, like LEA proteins; or (iii) enzymes required for the biosynthesis of osmoprotectants [10,18,48,49,50]. These changes finally lead to the adjustment of the plant’s metabolism [51,52].

### 2.4. Calcium-CaM and some Kinases Encoding Genes are Overexpressed by Cold Stress

At the end of the cold night (8 h), the functional category “signaling calcium” (Mapman BIN 30.3) was over-expressed (*p*-value-Mapman = 1.3 × 10^−3^, Table 1, Appendix A, Figure 3a). More precisely, the expression of genes encoding calmodulin (CaM) and proteins activated by CaM were up-regulated. These genes are known to be induced by low temperatures and involved in the cold signaling pathway [16,34,53,54]. In grapevine shoot tips exposed to chilling (5 °C), Tattersall et al. [33] reported that the expression of many calcium-signaling transcripts was affected. Moreover, these transcripts were more abundant than in responses to either salinity or osmotic stresses. Our results confirm the pivotal role of calcium/calmodulin-mediated signaling in plant responses to cold stress as suggested by Doherty et al. [55]. These authors described a mechanism whereby calcium increase, due to low temperature, leads to the formation of a calcium-CaM (or CaM-like protein)-CAMTA3 (a CaM-dependent transcriptional activator) complex that stimulates the transcription of *CBF1* and *CBF2*. The possible function of the calcium-CaM-CAMTA complex was also suggested in the DREB1/CBF signaling pathway [53].

The over-expression of BIN 30.3 slightly increased at 2 h PS (*p*-value-Mapman = 5.2 × 10^−4^, Table 2) with the induction of genes encoding CaM and proteins activated by CaM (Appendix A; Figure 3b). Two genes encoding CDPK (BIN 29.4) and calreticulin (CRT) were also up-regulated (Appendix A). The induction of *CRT* has been reported in response to a variety of stresses, such as cold [56]. Moreover, CRT and CDPK were reported to be associated with the development of tolerance to various stresses [57]. In our conditions, a Ca^2+^ dependent signal could also initiate the mitogen activated protein kinases phosphorylation cascade, as suggested by the over-expression of three genes encoding MAPK, Mitogen-Activated Protein Kinase Kinase Kinase 5 - MPKKK5 (BIN 30.6), and Mitogen-Activated Protein Kinase Kinase 2 - MAPKK2 (BIN 29.4) (VIT_15s0046g02010; VIT_18s0001g11240, VIT_11s0016g01770). At this time, the functional categories of “signaling receptor kinase wheat” LRK10 like (BIN 30.2.20; *p*-value = 1.8 × 10^−4^, Table 2) and “signaling receptor kinase misc” (BIN 30.2.99; *p*-value = 9.1 × 10^−4^, Table 2) were over-represented among the induced transcripts (Appendix A). These categories contain genes encoding receptor serine/threonine kinase and receptor-like protein kinase (RPK, Appendix A; Figure 3b). In plant cells, these receptors are known to respond to external challenges from the environment, including cold stress [2,58,59]. Moreover, the expression of three genes encoding serine/threonine protein kinase increased (Appendix A), including the cold-inducible kin3 protein (VIT_18s0122g01260). In plant cells, this kinase is known to convert information from receptors into appropriate signals to regulate many biological processes [60]. In our conditions, the expression of two genes encoding G-type lectin S-receptor-like serine threonine-protein kinase (Appendix A; Figure 3b) was also up-regulated at 2 h PS. The over-expression of its encoding genes provides salt-stress tolerance in *Glycine soja* [61]. This gene is also induced in response to ABA and drought stress [61]. Besides phosphorylation events, dephosphorylation processes mediated by protein phosphatase are also important events in the signaling pathways that regulate various cellular activities [62,63]. We showed the induction of genes encoding phosphatase 2C (BIN 29.4) at 2 h PS (Appendix A), which is in accordance with the works of Seki et al. [2] in *Arabidopsis*. This protein is known to play a prominent role in stress signaling in plants [64].

### 2.5. Transcription Factors

#### 2.5.1. Genes Encoding CBF Transcription Factors are Induced by Cold Stress

Potentially, cold stress induces the expression of seven *CBF* genes in grapevine [65,66,67], mostly in the leaf [17,68,69]. Our results show that the expression of two genes encoding DREB1/CBF proteins (VIT_02s0025g04460 and VIT_19s0138g00090) was up-regulated at 8 h (Appendix A). The expression profile of VIT_02s0025g04460 was validated by RT-qPCR and was found to be over-expressed at 8 h (fold change > 75, as compared to the control; Figure 2). The induction of *VvCBF2*, *VvCBF4*, and *VvCBFL* (an undefined CBF-like transcription factor) expression was also reported in the stem, tendril, flower, and berry skin treated at 4 °C [69]. The cold tolerance test in *Arabidopsis* over-expressing these genes showed that plants remained viable and noticeably healthy after exposure to cold treatment, suggesting that *VvCBF2*, *VvCBF4*, and *VvCBFL* may be responsible for conferring tolerance to low temperatures in grapevines without cold acclimation [69].

CBF protein contains a highly-conserved ERF/AP2 (Ethylene Responsive Factor/APETALA2) domain and belongs to a large multigene family of TFs widespread in the plant kingdom. These TFs are involved in the control of primary and secondary metabolism, growth as well as responses to environmental stimuli, including temperature stress [70], and may play fundamental roles in modifying the transcriptome under cold stress. Moreover, *CBF* genes are activators of transcription whose DNA binding domains are of the AP2/EREBP type (Ethylene-Responsive Element Binding Protein; [71,72]) and CBF proteins are defined as AP2 class transcription factors [73]. In our conditions, the functional category, “RNA.Regulation of transcription.AP2_EREBP” (BIN 27.3.3), was particularly over-represented (Appendix A, Figure 4 and Figure 5, *p*-value = 9.5 × 10^−13^, Table 1), which confirms Licausi et al.’s results [70] and the work of Xin et al., which reported an over-expression of this category in the grapevine leaf in response to cold stress [34].

#### 2.5.2. Cold Stress Affects the Expression of Genes Encoding Transcription Factors Involved in the Circadian Clock

In response to the cold night, the functional category concerning the TFs family of “Pseudo Arabidospsis thaliana response regulator (ARR)” (BIN 27.3.66) was over-expressed (*p*-value = 4.7 × 10^−7^) at 8 h (Table 1, Appendix A). This category contains *PSEUDO-RESPONSE REGULATOR* (*PRR*) genes, which are components of the circadian clock [74,75,76]. However, little information is available concerning *PRR* genes and their link with cold stress responses. In *Arabidopsis*, Nakamichi and coworkers reported that the attenuation of the PRR function obtained by mutations of the gene led to a noticeable cold tolerance due to DREB1/CBF-dependent pathway activation [74,75,76]. Genes encoding CBF1-3 were up-regulated and it was reported that PRR5 functions as a repressor, directly regulating CBF TFs. In our conditions, the expression of genes encoding PRR5 (Appendix A) was up-regulated, as well as DREB/CBF-encoding genes. These different responses may be due to the circadian disorder induced by mutations in *PRR* genes used in the works by Nakamichi’s team [74,75,76], or the time of the cold stress exposure (dark vs light).

The cold night also altered the expression of other TFs involved in the circadian clock. In particular, the functional categories of the MYB-related TF family (BIN 27.3.26, *p*-value = 6.6 × 10^−6^) and CO-like (Constans-like zinc finger family; BIN 27.3.7, *p*-value = 2.0 × 10^−8^) were down-regulated at 8 h (Table 1, Appendix A, Figure 4). These categories contain genes encoding late elongated hypocotyl (LHY, VIT_15s0048g02410) and protein reveille 1 (RVE1, VIT_04s0079g00410) MYB-related TFs, and CONSTANS-like proteins, CO1 and CO2 (VIT_14s0083g00640 and VIT_11s0052g01800). The overlap between stress- (including cold stress) and circadian-regulated gene expression has been highlighted in previous studies [77]. However, little information is available concerning the down-regulation of these genes in the context of cold stress responses, especially at night. Recently, Kinmonth-Schultz et al. [78] reported that cool temperature treatments at night (17 °C or 12 °C) increased CONSTANS transcript levels in the frame of flowering process regulation in Arabidopsis. Regarding RVE1, it has been reported that high freezing tolerance levels of Arabidopsis lines coincided with down-regulation of this gene, supporting the hypothesis of RVE1 being a negative regulator of freezing tolerance [79]. Our results obtained in the grapevine flower are in agreement with these reports, with the noticeable exception of the *CO* genes.

#### 2.5.3. Heat-Shock Factors

As mentioned above, plants exploit ROS as signaling molecules and plant cells sense ROS via redox-sensitive TFs, such as nitrogen permease reactivator or heat-shock factors, which activate functional proteins involved in the re-establishment of cellular homeostasis [80]. In our study, two genes encoding heat-shock factors were up-regulated at the end of the cold night (VIT_09s0018g01320 and VIT_07s0031g00670; Appendix A) as already reported in *Arabidopsis* under short cold stress [81].

#### 2.5.4. Cold Impacts the Expression of EDS1, SGT1, SAG101, TIR-NB-LRR, PHI-1, Genes Reported to be Involved in Global Stress Responses

Finally, our data suggest that other genes could be involved in stress signaling. These genes are classified in the over-expressed functional category of “stress.biotic.signaling” (BIN 20.1.3; *p*-value = 3.0 × 10^−8^, Table 2), namely EDS1 (enhanced disease susceptibility 1), SGT1 (suppressor of G2 Allele SKP1), and Mildew resistance Locus O (MLO) proteins (Appendix A). The search for interacting partners for EDS1 in the context of basal resistance to plant pathogens has gained much attention in recent last years. Among these partners are (i) PAD4 (Phytoalexin Deficient 4) and SAG101 (Senescence-Associated Gene 101) proteins, (ii) TIR-NB-LRR (Toll-interleukin-1 receptor-nucleotide binding-leucine rich repeat) resistance proteins, and (iii) salicylic acid (SA) [82,83,84]. Nevertheless, this regulatory hub is also implied in abiotic stress responses, and the importance of EDS1 as a central regulatory protein in oxidative stress signaling has been proposed [84,85]. In the grapevine flower, we showed that the expression of genes encoding EDS1, SAG101, and TIR-NB-LRR was up-regulated at 2 h PS (Appendix A), as well as the expression of genes implicated in salicylic acid synthesis/degradation (BIN 17.8.1; *p*-value = 7.4 × 10^−4^, Table 2, Appendix A). Our data also highlighted the induction of genes encoding SGT1 at 2 h PS (VIT_16s0050g01510; VIT_12s0035g01560). This up-regulation is in accordance with recent results obtained in *Brassica oleracea* [86]. Thus, we can suggest that the EDS1/SGT1 regulatory hub is involved in the response to a cold night in the grapevine flower.

The functional category of “signaling.in sugar and nutrient physiology” (BIN 30.1) was also found to be highly over-expressed in the grapevine flower at 8 h (*p*-value = 5.8 × 10^−5^, Table 1, Appendix A, Figure 3a), in particular three genes encoding phosphate-induced protein (Phi-1). The exact function of this protein is not known, but its implication in multiple abiotic stress responses was reported [33,87]. In grapevines, *PHI-1* was more highly expressed in response to cold and its expression may also be related to the increase in phosphate ions observed at 8 h in response to cold [33,34]. Moreover, it was reported that *PHI-1* is involved in calcium signaling and would seem to be CDPK-specific [88].

### 2.6. Cold Modulates the Expression of Genes Involved in Global Stress Responses

#### 2.6.1. Cold Night Induces an Over-Expression of Genes Encoding Chaperone Proteins

At 8 h, the expression of many genes coding for HSPs, also identified as molecular chaperones with strong cryoprotective effects [2,12,89,90], was up-regulated (in particular HSP 70 and 101) (Appendix A). Indeed, we showed that functional category of “stress.abiotic.heat” (BIN 20.2.1) was over-expressed at this time (*p*-value = 3.1 × 10^−7^, Table 1), but was down-regulated at 2 h PS (*p*-value = 1.7 × 10^−7^, Table 2). Moreover, it appears that the functional category of “stress.biotic.PR-proteins” (BIN 20.1.7) was over-expressed (*p*-value = 6.1 × 10^−3^ at 8 h and 4.2 × 10^−3^ at 2 h PS) (Table 2, Appendix A, Figure 4). Some pathogen-related (PR) proteins are also responsive to low-temperature stress, since these proteins have antifreeze activity [10,89,91]. In our conditions, the expression of some genes encoding PR proteins (SCUTL2, Class IV chitinase, and others: Appendix A) was up-regulated after cold stress. The expression profiles of VIT_05s0094g00360 encoding Class IV chitinase (CHI4C), of VIT_203s0088g00810 encoding a pathogenesis-related protein 1 (PR1), and of VIT_13s0064g01310 encoding a thaumatin like protein (TL1) were validated by RT-qPCR (Figure 2). Thus, the transcript level of *CHI4C* was approximately 3.5 fold higher at 2h PS and *PR1* exhibited an induction at 8 h (fold change > 5). Nevertheless, *TL1* expression was strongly downregulated (5-fold less) at 2 h PS (Figure 2). Moreover, the expression of three genes encoding PR-10 protein was down-regulated after cold night exposure (VIT_05s0077g01540; VIT_05s0077g01550; VIT_05s0077g01580; data not shown). The classification of this protein is not well known yet (see Fernandes et al. [92] for a review). According to our results, if some of the genes encoding PR-proteins seem to be involved in the grapevine flower’s response to cold stress, *TL1* and *PR-10* do not seem to be implicated, different from what was observed in the leaf and roots of *Tamarix hispida* [93] and ginseng plantlets [94].

Like ERF, CBF targets are described as cold-inducible genes (*COR*). *COR* genes include those encoding LEA proteins and, more globally, dehydrins. LEA proteins represent a major class of cryoprotective molecules [95,96,97] thought to be important for membrane stabilization and the protection of proteins from denaturation. LEA and the dehydrins ERD (early response to dehydration), known to function as chaperones, were reported to accumulate during cold stress [98,99]. In agreement with these functions, in our conditions, we showed that the expression of one gene encoding ERD (VIT_10s0116g00940; Appendix A) was up-regulated at 8 h. At 2 h PS, the up-regulation of this gene persisted and the expression of another gene encoding ERD (VIT_10s0003g04180), as well as the expression of two genes encoding LEA (VIT_15s0046g02090; VIT_15s0046g02110) were also up-regulated (Appendix A).

In our conditions, we showed that the functional category of “hormone metabolism.abscisic acid.induced-regulated.responsive-activated” (BIN 17.1.3) was also over-expressed at 2 h PS (*p*-value = 1.6 × 10^−3^, Table 2, Appendix A). In this category, we find, notably, genes encoding Hva22 protein. In barley, it was reported that the expression of *HVA22* is induced by cold stress and a function of this protein in cell protection was suggested [100]. This protective function has been confirmed recently by heterologous expression in *Escherichia coli* [101]. Moreover, it was reported that *HVA22* shares homology with other ABA-responsive genes, such as *LEA* [100]. Thus, we suggest that this protein may be involved in cell’s protection of the grapevine flower in response to cold stress.

#### 2.6.2. Transcripts Involved in Detoxification Pathways are Induced by Cold Stress in Grapevine Flowers

Plants possess very efficient enzymatic antioxidant defense systems, which regulate the cascades of uncontrolled oxidation and therefore protect plant cells from oxidative damage. Several enzymes, such as glutathione-*S*-transferase (GST) and peroxidase, are involved in this detoxification system. In the present study, we showed that the functional categories of “misc.glutathione *S* transferases” (BIN 26.9) and “misc.peroxidases” (BIN 26.12) were over-expressed at 2 h PS (*p*-value = 1.5 × 10^−3^ and 2.1 × 10^−3^, respectively, Table 2, Appendix A, Figure 4b). The over-expression of genes encoding these proteins has already been reported under cold stress [2]. In grapevines, the over-expression of genes encoding GST is namely linked with cold resistance [102]. So, it seems that this antioxidant defense system is active in the grapevine flower in response to a cold night. Peroxidases have also been proposed to be involved in lignin biosynthesis. In particular, we noticed the over-expression of the gene encoding peroxidase, ATPA2 (VIT_18s0001g13110; Appendix A; Figure 4b), involved in the lignification process in *Arabidopsis* [103]. Thus, we suggest that lignification is activated in the cold stress response by strengthening the cell walls in the grapevine (see Section 2.7.2 cell wall part).

### 2.7. Metabolisms Impaired by the Cold Night

#### 2.7.1. Carbon Metabolism

Cold stress impacts the expression of genes encoding proteins of the photosystem antenna complexes.

The expression of several genes coding for chlorophyll a/b binding proteins, light-harvesting chlorophyll binding protein precursor, photosystem I (PSI) subunits, and the reaction center were repressed, with a marked impact 2 h PS (Appendix A, Figure 5b and Figure 6b). The functional category of “PS.lightreaction.photosystem II.LHC-II” (BIN 1.1.1.1) was particularly down-regulated 2 h PS (*p*-value = 4.8 × 10^−15^, Table 2). The expression profiles of VIT_00s0181g00180 and VIT_17s0000g06350 encoding chlorophyll a/b-binding proteins were validated with RT-qPCR. The expression of these two genes was down-regulated at 2 h PS (Figure 2). The light-harvesting chlorophyll a/b binding (LHCB) proteins are, with chlorophyll and xanthophylls, the apoproteins of the photosystem II (PSII) complex [104]. The expression of *LHCB* genes is modified by multiple environmental stresses, such as cold [105], and play an important role in plant adaptation [106,107,108]. The regulation of their expression is also considered as an important mechanism to modulate chloroplast functions [109]. Under low temperatures, the absorbed light energy cannot be used productively and such an energy imbalance leads to an over-excitation of the photosynthetic apparatus that in turn increases the potential for photoinhibition [110] and, subsequently, photooxidative damage [111]. Thus, it appears that in grapevine flowers subjected to cold stress, there is a down-regulation of the genes associated with photosynthetic light harvesting in response to cold stress, in order to reduce the light energy absorbed and to avoid an energy imbalance.

At 2 h PS, the expression of genes encoding PSII cytochrome b559 (cyt b559; VIT_13s0101g00200) and NADH-plastoquinone oxidoreductase subunit K (VIT_07s0031g01060) was slightly induced (Appendix A; Figure 5b and Figure 6b). Cyt b559 is an integral component of the PSII complex [112], involved in the cyclic electron transfer around this photosystem, notably as a protective mechanism [113]. NADH-plastoquinone oxidoreductase subunit K belongs to the the NADH dehydrogenase (NDH) complex, which works as an electron transport intermediate in the cyclic flow around PSI [114]. This flow contributes to the generation and maintenance of the transmembrane pH gradient and is essential for efficient photosynthesis [115], but also photoprotection [116,117]. So, we can suppose that a protection of the photosynthetic chain against an energy excess is set up, which is in accordance with the disturbance in Pinot Noir photosynthetic chain activity observed following a cold night [29].

In our conditions, we noticed an over-expression of genes encoding a large subunit of ribulose-1,5-bisphosphate carboxylase/oxygenase 2 h PS (Rubisco; BIN 1.3.1 “PS.calvin cycle.rubisco large subunit”; *p*-value = 2.4 × 10^−5^; Appendix A; Figure 5b and Figure 6b). These results are in agreement with a previous work on the grapevine leaf after 8 h of cold stress in the light [34]. At least, it seems that photosynthetic CO_2_ assimilation and/or photorespiration were not inhibited after the cold night.

Expression of genes involved in synthesis of osmoprotectant sugars are differently impacted during stress and recovery

At the end of the cold night, the functional category of “minor CHO metabolism.raffinose family.galactinol synthases” (BIN 3.1.1.2) was particularly down-regulated (*p*-value = 2.5. × 10^−6^, Table 1) and then up-regulated (*p*-value = 7.6 × 10^−6^, Table 2) during the recovery period (Appendix A; Figure 7). In stressed grape berries, the induction of genes coding for galactinol synthase (*GOLS*) was already reported, notably during the day [118,119]. Galactinol synthase catalyzes the first step in the biosynthesis of the raffinose family oligosaccharides (RFOs). RFOs protect plant cells from the oxidative damage generated by various stress conditions [120,121,122]. Other roles have been proposed for RFOs, including ROS scavenging, protection from photoinhibition, and also metabolic detoxification [123]. Galactinol synthase also plays a key regulatory function in the carbon partitioning between sucrose and RFOs [124], and is therefore a potential metabolic control point to manage the levels of RFOs. Then, raffinose synthesis is catalyzed by the raffinose synthase from sucrose and galactinol. In maize seeds, it was suggested that the ratio of sucrose to RFO is crucial to desiccation tolerance rather than the total amount of sugars [125]. So, we hypothesize that the down-regulation of *GOLS* at 8 h was due to the maintenance of a high sucrose level. It is known that the accumulation of sucrose, as well as other simple soluble sugars, contributes to the stabilization of biological membranes [4,43]. The induction of genes coding for raffinose synthase (RS; VIT_14s0066g00810) and alkaline alpha galactosidase II (VIT_19s0015g01350), involved in raffinose synthesis, was measured after the cold night (Appendix A), which is coherent with their role as an osmoprotectant [2,126].

We also observed the down-regulation of two vacuolar invertases (*GIN1* and *GIN2*; VIT_16s0022g00670 and VIT_02s0154g00090) (BIN 2.2.1.3.3 “major CHO metabolism.degradation. sucrose.invertases.vacuolar”; *p*-value = 2.6 × 10^−4^, Table 2). The expression profiles of genes encoding RS and GIN1 were validated in RT-qPCR (Figure 2). Thus, at 2h PS, the transcript level of RS was 4 folds higher, while GIN1 expression was decreased (around 3 fold). So, it seems that raffinose synthesis was induced at 2 h PS as already demonstrated in the leaf and crown of barley under cold stress [127]. The inhibition of sucrose degradation could thus provide the necessary sucrose ratio. The expression of a gene encoding β-amylase1 (VIT_05s0077g00280), involved in starch degradation, was repressed at 8 h, in agreement with the starch accumulation observed in Pinot noir flowers at this time [29]. Thereafter, the starch synthesis functional category (BIN 2.1.2) was particularly down-regulated at 2 h PS (*p*-value = 4.1 × 10^−3^, Table 2) with the repression of granule-bound starch synthase (VIT_02s0025g02790) and soluble starch synthase (VIT_10s0116g01730) expression. At this time, the expression of genes encoding β-amylase 3 (VIT_02s0012g00170) was induced, while the expression of two genes encoding AGPase (ADP-glucose pyrophosphorylase) was repressed (VIT_18s0089g00190 and VIT_12s0057g01480) (Appendix A; Figure 5b). Consequently, these data suggest that the cold night could have induced an accumulation of maltose and glucose in the grapevine flower, as reported in the grapevine leaf [34] and the crown of barley [127] in response to cold stress. Maltose acts as a cryoprotectant [128] and represents a precursor for soluble sugar metabolism [128,129]. So, its accumulation during the cold night could provide better flower protection.

We noticed an over-expression of *SUC27*, a gene encoding a sucrose transporter, at 2h PS (VIT_18s0076g00250; Appendix A). Its expression profile was validated with RT-qPCR (Figure 2). This result agrees with that obtained in the barley crown under cold stress [127]. This up-regulation suggests a sucrose import in the flower in response to the cold night, which is in correlation with the increase in hexoses and sucrose concentrations in cold-stressed inflorescences [27].

#### 2.7.2. Cold Modulates the Expression of Genes Encoding Components Involved in Cell Wall Degradation and Synthesis

Cell wall-related proteins, which include many enzymes, such as xyloglucan endo-β-transglycosylases/hydrolases, endo-1,4-β-D-glucanase, expansins, pectin methylesterase, polygalacturonase, pectin/pectate lyase-like, and pectin acetylesterase, play a central role in modulating cell wall extensibility (anisotropic and isotropic expansion) and plasticity [130,131,132]. These enzymes may consequently contribute to the adjustment to environmental stresses. In response to the cold night, cell wall degradation, namely concerning the pectate lyases and polygalacturonases (BIN 10.6.3) category, was particularly down-regulated at 8 h and 2 h PS (*p*-value = 1.4 × 10^−5^ and 1.7 × 10^−7^, respectively; Table 1 and Table 2, Appendix A; Figure 5). This phenomenon has also been observed in the grapevine leaf in response to cold stress [34]. Polygalacturonase is one of the hydrolases responsible for cell wall pectin degradation [133]. In young rice leaves, it was reported that the over-expression of a gene encoding a PG1 subunit reduces cell adhesion caused by pectin degradation, leading to an increase in abiotic stress sensitivity [133]. Polygalacturonases also partake in the plant’s organ abscission phenomenon [134]. Woody plants have developed abscission to facilitate the shedding of no longer needed, infected, damaged, or senescent vegetative or/and reproductive organs [135,136,137]. Modifications in the expression of cell wall hydrolytic enzymes are one of the most frequent reported phenomena during abscission [138,139]. Indeed, cell wall modifying enzymes are over-expressed in the cell separation zones of several organs, including pedicel [140], flowers, and floral parts [141]. In grapevines, the flower and fruit abscission process leads to important yield losses when enhanced by stresses [142] depending on cultivars. Pinot noir has been found to be relatively tolerant under environmental stress [142]. We suggest that the down-regulation of the expression of genes encoding PG may contribute to this better tolerance.

The expression profile of one gene encoding PG (VIT_01s0127g00400) was validated with RT-qPCR. The transcript level of PG decreased approximately 4 times at 2 h PS (Figure 2). At 2 h PS, we showed that functional category of “cell wall.cellulose synthesis.cellulose synthase” (BIN 10.2.1) was also down-regulated (*p*-value = 1.7 × 10^−6^; Table 2, Appendix A). So, it seems that in our conditions, cell wall degradation and synthesis stopped in response to the cold stress, similarly to cotton [143].

Secondary cell walls contain a wide range of additional compounds, such as lignin. Lignification and changes in lignin content can occur during growth at low temperatures [144] and can be enhanced during cold acclimation and participate in cell wall modification by strengthening it [131]. In our conditions, we showed that the functional category of “secondary metabolism.phenylpropanoid.lignin biosynthesis.CAD/SAD” (Cinnamyl/Sinapyl Alcohol Dehydrogenase; BIN 16.2.1.10) was particularly up-regulated at 2 h PS (*p*-value = 5.3 × 10^−6^, Table 2, Appendix A). Because of its specific role in the monolignol biosynthetic pathway, *CAD* expression is considered as a marker of lignin biosynthesis [145]. Under cold stress, it was already reported that *CAD* gene expression was induced in the sweet potato and *Arabidopsis* [2,146]. Moreover, it has been proposed that cell wall rigidity may be an important factor in cell resistance to freeze-induced dehydration [147]. Cell wall rigidity affects the pore size and decreases it, as shown in an apple and grape cell suspension during cold acclimation: By limiting the pore size, ice can be excluded from the cell [147]. It is, however, also true that, more recently, a decrease in lignification in the *TCF1* thale cress mutant was correlated with an enhanced frost tolerance, since this results in a less compact cell wall structure that allows the growth of ice crystals [148]. The expression of other genes involved in lignin biosynthesis, including 4-coumarate:CoA ligase (*4CL*; VIT_01s0010g03720) and Caffeoyl-CoA *O*-methyltransferase (*CCoAOMT*), was up-regulated in response to the cold night (Appendix A), which is in agreement with results obtained in the pea and *Phaseolus vulgaris* [149,150]. The expression profile of a gene encoding 4CL was validated by RT-qPCR. The level of transcripts increased more than 4 and 10 fold at 8 h and 2 h PS, respectively (Figure 2). In our conditions, the expression of two genes encoding laccase-4 (VIT_06s0004g06090 and VIT_09s0018g00950) was particularly down-regulated (Appendix A). Laccases are implied in monolignol polymerization, but may also have roles in the polymerization of other secondary metabolites originating from the phenylpropanoid pathway, cell wall chemistry, or integrity [151,152]. So, the down-expression of these genes can seem inappropriate. However, it was recently shown in the *Arabidopsis* stem that the disruption of LAC4 only leads to a relatively small change in lignin content and only under continuous illumination [152]. A redundancy in terms of functions is indeed present for the laccase multigene family in thale cress. Finally, we showed that the expression of other genes of this pathway was down-regulated at 2 h PS (Appendix A), particularly, genes encoding phenylalanine ammonia lyase (PAL; BIN 16.2.1.1; *p*-value = 3.0 × 10^−20^, Table 2). The down-regulation of genes encoding PAL has also been observed in the barley crown following exposure to cold [127]. Nevertheless, the results obtained concerning the response of the wheat crown to cold stress showed that the lignin content was increased, while PAL activity was reduced [153]. Therefore, since lignin and its precursors are important components of the cell wall, any change in their content and/or composition is bound to have an impact on its physical and mechanical properties. Consequently, in light of the data obtained, it is reasonable to assume that the lignification process is engaged in the response to a cold night by the grapevine flower.

Am analysis using Cytoscape (showing only pathways with *p*-value < 0.05; *p*-value correction with the test of Benjamini-Hochberg) performed by ranking the up- and down-regulated genes at 2 h (red and green, respectively) and the up- and down-regulated ones at 8 h (blue and orange, respectively) showed an enrichment of the ontology related to sporopollenin biosynthesis and pollen exine formation (Appendix A). This is interesting because, despite the lack of knowledge concerning sporopollenin composition [154], it is known that it contains aromatic moieties derived from the phenylpropanoid pathway. The increased expression of genes involved in the phenylpropanoid pathway may thus be linked to sporopollenin formation. The ClueGO analysis [155] revealed that four genes are found in the sporopollenin biosynthetic GO: VIT_01s0010g03720, VIT_03s0038g01460, VIT_03s0038g04220, and VIT_15s0046g00330. VIT_01s0010g03720 is the homologue of the *Arabidopsis* ACYL-COA SYNTHETASE 5 (ACOS5) [156], which together with POLYKETIDE SYNTHASE B (PKSB, the *Vitis* homologue is VIT_03s0038g01460), TETRAKETIDE ALPHA-PYRONE REDUCTASE 1 (TKPR1, the *Vitis* homologue is VIT_03s0038g04220), and PKSA colocalize in the ER and form a metabolon. VIT_15s0046g00330 is the homologue of CYP703, which was shown to interact with ACOS5 in a yeast two-hybrid assay [154].

#### 2.7.3. Secondary Metabolism

According to several studies, the expression of secondary metabolism genes, namely those implied in the biosynthesis of flavonoids, anthocyanins, and terpenoids, is generally well correlated with cold tolerance [9,10,11,157].

Cold down-regulates the expression of genes related to the flavonoid pathway

In our conditions, the cold night involved the down-regulation of the expression of some genes implied in anthocyanins biosynthesis (8 h), as well as genes encoding chalcone synthase (2 h PS; Appendix A). Nevertheless, at 2 h PS, we showed the up-regulation of the expression of two genes encoding stilbene synthase (STS and STSa; VIT_16s0100g00780 and VIT_10s0042g00840). This result was confirmed by RT-qPCR (x4.2 and 6.3, respectively; Figure 2). So, we hypothesize that, in our conditions, flavonoid biosynthesis is down-regulated (including genes encoding PAL, acting upstream of the pathway) to shunt the precursors towards the biosynthesis of stilbenoids, since these enzymes utilize the same substrates and catalyze the same condensing type of enzyme reaction (Figure 8). The up-regulation of *STS* has been reported in grape species in response to cold stress [34,47,158], which potentially links the biotic defense and antioxidant function of resveratrol with damage to plant cells from low temperatures. Finally, we showed that the functional category of “secondary metabolism.flavonoids.chalcone.pterostilbene synthase” (BIN 16.8.2.3.1) was down-regulated at 2 h PS (*p*-value = 4.1 × 10^−6^, Table 2, Appendix A), which seems indicate that pterostilbene is not a form of resveratrol synthesized in response to a cold night in this organ.

The expression of valencene synthase encoding gene is down-regulated by the cold night

In response to the cold night, we showed that the expression of several genes involved in terpene metabolism (BIN 16.1.5) was particularly down-regulated, namely genes encoding valencene synthase, germacrene synthase, and beta-caryophyllene synthase (Appendix A; Figure 5). To the best of our knowledge, this response has not been described so far and we hypothesize that in the flower, this pathway is down-regulated to compensate for the lack of energy production and reduction of carbon assimilation by the photosynthetic chain.

## 3. Materials and Methods

### 3.1. Plant Materials and Cold Treatment

Fruiting cuttings have a similar reproductive physiology to vines, and can consequently be used as a reliable material to study grape physiology [160]. Fruiting cuttings of Pinot noir were obtained from grapevine canes according to the protocol of Sawicki et al. [29]. Experiments were carried out using cuttings with four leaves harboring inflorescences at the female meiosis stage (corresponding to BBCH55 + 2 days stage; adapted to [160]). Control plants were maintained in a growth chamber for 8 h at 19 °C (control night). For the cold-treatment, plants were placed at 0 °C for 8 h. Sampling was then done at the end of the cold night (8 h) and 2 h after the end of the cold-night (2 h post-stress – 2h PS). For the latter, control and stressed plants were collected after 2 h at 25 °C in the light. Six inflorescences, sampled from 6 different plants, were used for each treatment and each time point.

### 3.2. RNA Extraction

For the biological repetitions and each point, RNA samples were obtained by pooling RNAs from 6 inflorescences (from 6 different plants). Samples were ground with liquid nitrogen and 100 mg of powder was used for the total RNAs extraction. To obtain highly pure RNAs, we used a new protocol using Plant RNA Purification Reagent (Invitrogen, Cergy Pontoise, France) associated with purification on a Macherey-Nagel column. Briefly, the aqueous phase obtained after RNA extraction using Plant RNA Purification Reagent was recovered and filtered on the column according to the manufacturer’s instructions. Highly pure RNAs were finally eluted in 30 µL of RNase-free water.

### 3.3. Microarray Analysis

Microarray analysis was carried out at the Institute of Plant Sciences Paris-Saclay (IPS2, Saclay, France), using the *Vitis* array based on Roche-NimbleGen technology. A single high-density *Vitis* microarray slide contains twelve chambers, each containing 145,869 primers representing 29,549 *Vitis vinifera* genes. Two independent biological replicates were produced. For each comparison, one technical replicate with fluorochrome swap was performed (i.e., 4 dye-switch hybridizations per comparison). Starting with 200 ng of total RNA, amplification and cDNA labeling with Cy3-dUTP or Cy5-dUTP (Perkin-Elmer-NEN Life Science Products) were performed with a TransPlex Complete Whole Transcriptome Amplification Kit (WT2A) (Sigma-Aldrich, St Louis, MO, USA) according to the manufacturer’s recommendations. The hybridization and washing were performed according to NimbleGen instructions (Roche NimbleGenR Technologies, Lyon, France). Two-micron scanning was performed with a InnoScan900 scanner (Innopsys, Carbonne, France) and raw data were extracted using Mapix software (Innopsys, Carbonne, France).

### 3.4. Statistical Analysis of Microarray Data

For each array, the raw data comprised the logarithm of the median feature pixel intensity at wavelengths of 635 nm (red) and 532 nm (green). For each array, a global intensity-dependent normalization using the loess procedure [161] was performed to correct the dye bias. The differential analysis was based on the log-ratios averaged over the duplicate probes and over the technical replicates. Hence, the numbers of available data for each gene equals the number of biological replicates and were used to apply a moderated t-test [162].

Under the null hypothesis, no evidence that the specific variances vary between probes was highlighted by the Limma package and consequently the moderated t-statistics was assumed to follow a standard normal distribution. To control the false discovery rate, adjusted *p*-values found using the optimized FDR approach [163] were calculated. We considered these as being differentially expressed genes corresponding to the probes with an adjusted *p*-value ≤ 0.05. was been used to smooth the specific variances by computing empirical Bayes posterior means. The library kerfdr was used to calculate the adjusted *p*-values.

### 3.5. Data Deposition

Microarray data from this article were deposited at Gene Expression Omnibus (http://www.ncbi.nlm.nih.gov/geo/), accession no. GSE87296 and at CATdb (http://urgv.evry.inra.fr/CATdb/; Project: 12plex_vitis_2012_04) according to the “Minimum Information About a Microarray Experiment” standards.

### 3.6. Transcriptome Analysis

Gene expression was evaluated using the number of differentially expressed genes (DEGs), clustering, and annotation. Differentially expressed genes were identified by using the Blast2GO software (Blast2Go 4.1.9, a research tool designed to enable Gene Ontology (GO)). Using this software, we performed gene annotations from the NCBI database. Global data analyses were performed and are summarized in Figure 9.

### 3.7. Functional Analyses

Functional analysis of the grapevine genes differentially expressed in response to a cold night was carried out with the Mapman software (MapMan 3.6.0 RC1, Max Plack Institue of Molecular Plant Physiology, Golm, Germany), which offers improved ontology and the ability to visualize gene expression overlaid into biochemical pathways or diagrams [164]. Mapman ontology terms’ over-representation analyses were performed using our mapping via a Bonferroni-corrected Fisher’s Exact Test (Upton, 1992) implemented in the CorTo software (available at http://www.usadellab.org/cms/index.php?page=corto).

### 3.8. Real-Time PCR Analysis

Microarray gene expression data were validated by real-time PCR using gene-specific primer pairs (Appendix A). Total RNAs (150 ng) were reverse-transcribed using a Verso cDNA Synthesis Kit (Thermo Fisher Scientific, Waltham, MA, USA) according to the manufacturer’s protocol. Real-time PCR was performed using Absolute Blue qPCR SYBR Green (Thermo Fisher Scientific, Waltham, MA, USA), in a CFX96 real-time PCR detection system (Bio-Rad, Hercules, CA, USA). The thermal profile was 10 s at 95 °C (denaturation) and 30 s at 60 °C (annealing/extension) for 40 cycles. The specificity of PCR amplification was checked using a heat dissociation curve from 65 to 95 °C following the final cycle. The PCR efficiency of the primer sets was calculated by performing real-time PCR on serial dilutions of cDNA. For each experiment, PCR reactions were performed in duplicate, and three independent experiments were analyzed. Results were normalized with two reference genes (EF1α and 60RSP, Appendix A), and the relative gene expression was determined with the formula, 2^−ΔΔ*C*t^, using CFX Manager 3.0 software (Bio-Rad, Hercules, CA, USA) (Figure 2).

## 4. Conclusions

Our microarray analysis provides the first results on the global transcriptomic pattern of the grapevine flower after cold stress. Our results confirm the presence of cold-responsive regulatory mechanisms described previously in other plants, namely:

(1) The pivotal role of calcium/CaM-mediated signaling of the over-expression of diverse transcription factors, such as CBF, ERF, or NAC.

(2) The induction of some genes encoding proteins known to function as chaperones (for instance *GST*, *HSP*, or *PR* proteins), sugar transporters, and enzymes involved in sugar synthesis, or genes involved in lignin biosynthesis.

(3) The down-regulation of genes encoding GOLS, pectate lyases, and polygalacturonases. However, this study brings new insight into the behavior of grape flowers in response to cold by also identifying novel mechanisms that were previously not known for their role in the cold stress response.

To summarize, we report:

(1) The up-regulation of genes encoding G-type lectin S-receptor-like serine threonine-protein kinase, PRR5, heat-shock factors, and the suppressor of the G2 allele, *SKP1*.

(2) The down-regulation of *MYB*-related transcription factors, Constans-like zinc finger family, and *PR10* genes.

Consequently, our results reveal interesting features and potentially valuable traits associated with the stress responses in the grapevine flower. The main results, summarized in Figure 10, constitute a global overview of the main functional gene categories that are significantly impacted by cold stress, and provide useful starting points for future investigations on cold exposure in early flower development. This is particularly relevant in the frame of the ongoing climate changes, with a general increase in temperatures that impacts grapevine phenology causing earlier bud break in the season, thus exposing young inflorescences to cold injury during spring.

## Figures and Tables

**Figure 1 ijms-20-01130-f001:**
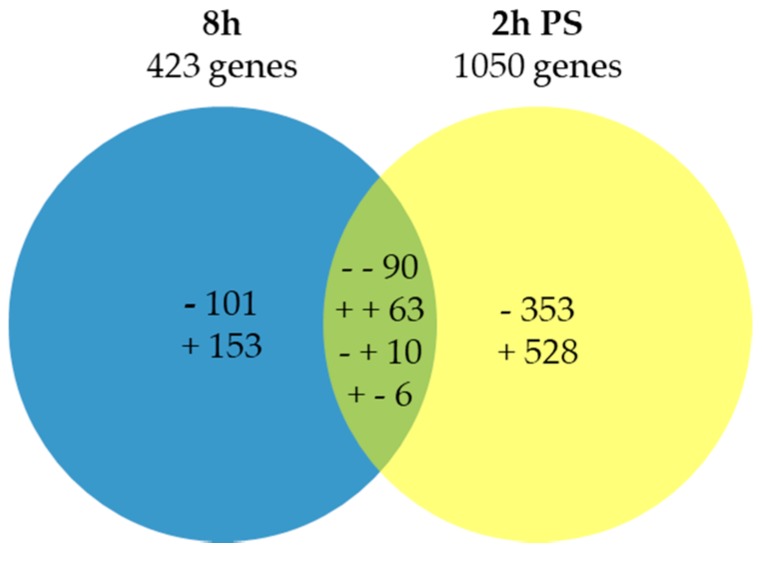
Venn diagram of the number of transcripts (both identified and unknown) that were up- or down-regulated by cold stress based on microarray analyses. The « + » and « - » indicate up- and down-regulated transcripts, respectively. In total, 423 and 1050 stress-inducible genes were identified at 8 h and 2 h PS, respectively. 101: unique down-regulated transcripts at 8 h; 353: unique down-regulated at 2 h PS; 153: unique up-regulated transcripts at 8 h; 528: unique up-regulated transcripts at 2 h PS; 90: commonly down-regulated transcripts at both time; 63: commonly up-regulated transcripts at both time; 10: down-regulated at 8 h but up-regulated at 2 h PS; 6: up-regulated at 8 h but down-regulated at 2 h PS.

**Figure 2 ijms-20-01130-f002:**
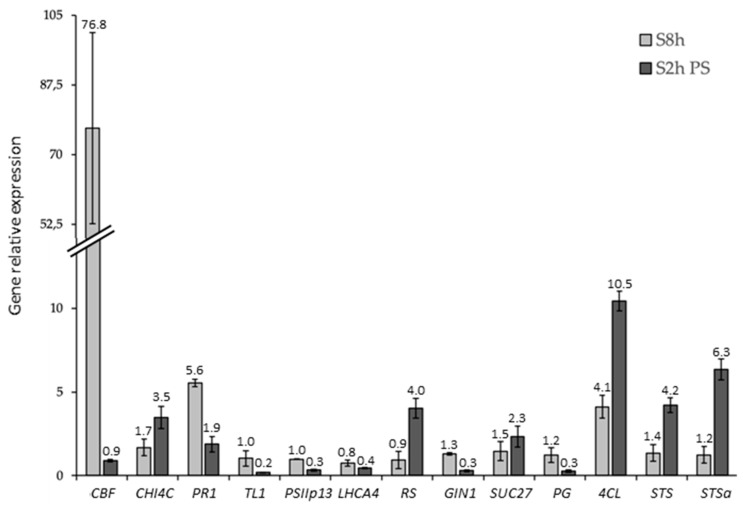
Microarray analysis validation by RT-qPCR. The expression levels of 15 selected genes were quantified by RT-qPCR at 8 h and 2 h PS. The y axis indicates the fold change in gene expression relative to the control plant at T8 h and T2 h PS. The expression profiles were similar by RT-qPCR and microarrays analysis. The data were obtained from three independent biological replicates and two technical repetitions. The numbers above the bars indicate the average fold change values with respect to the control conditions.

**Figure 3 ijms-20-01130-f003:**
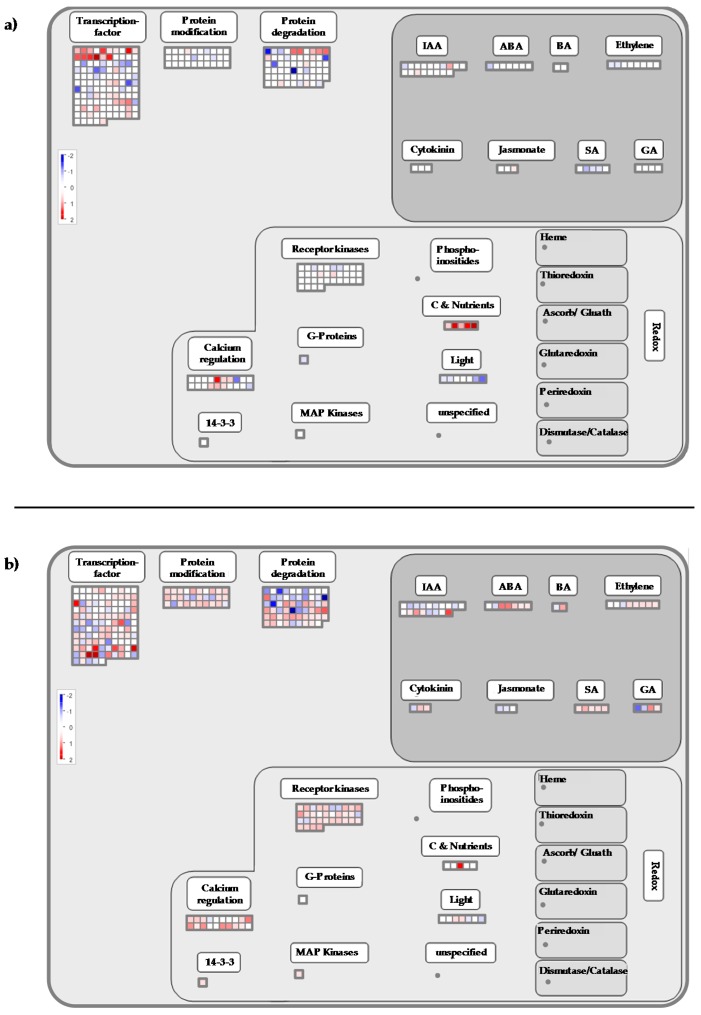
Mapman visualization of the regulation process pathway modulation at (**a**) 8 h and (**b**) 2 h PS.

**Figure 4 ijms-20-01130-f004:**
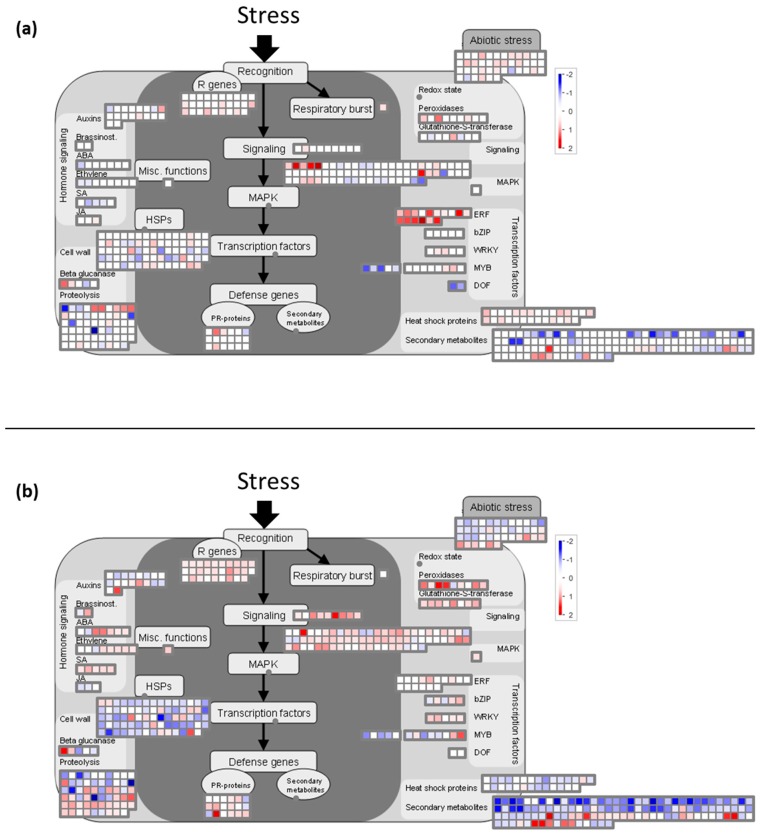
Mapman visualization of the stress pathway modulation at (**a**) 8 h and (**b**) 2 h PS.

**Figure 5 ijms-20-01130-f005:**
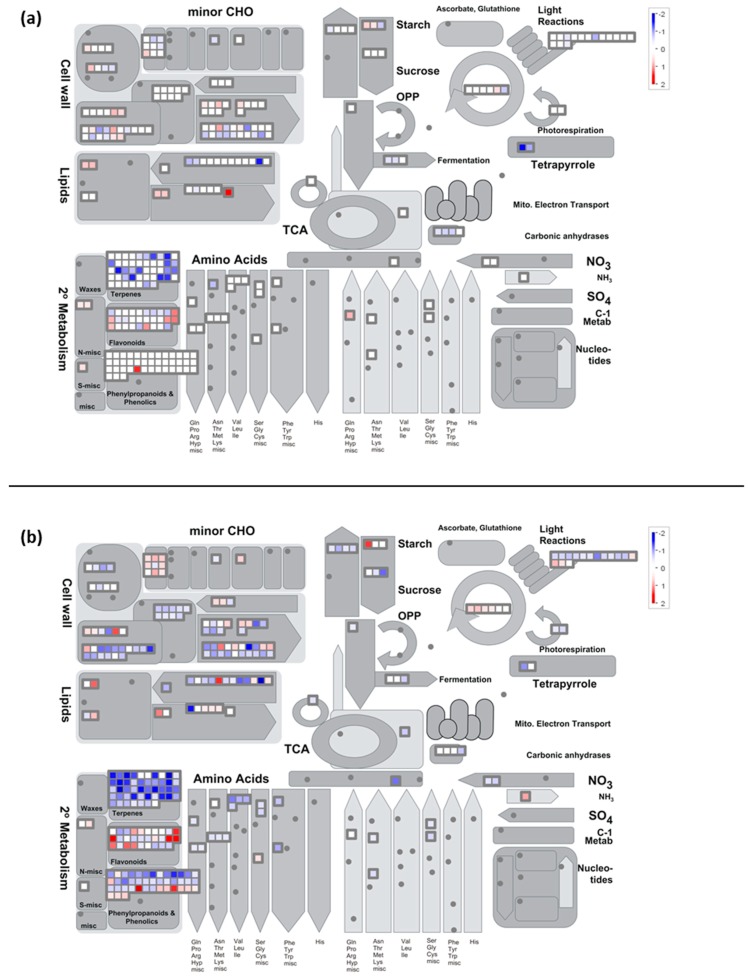
Mapman visualization of the metabolism overview pathway modulation at (**a**) 8 h and (**b**) 2 h PS.

**Figure 6 ijms-20-01130-f006:**
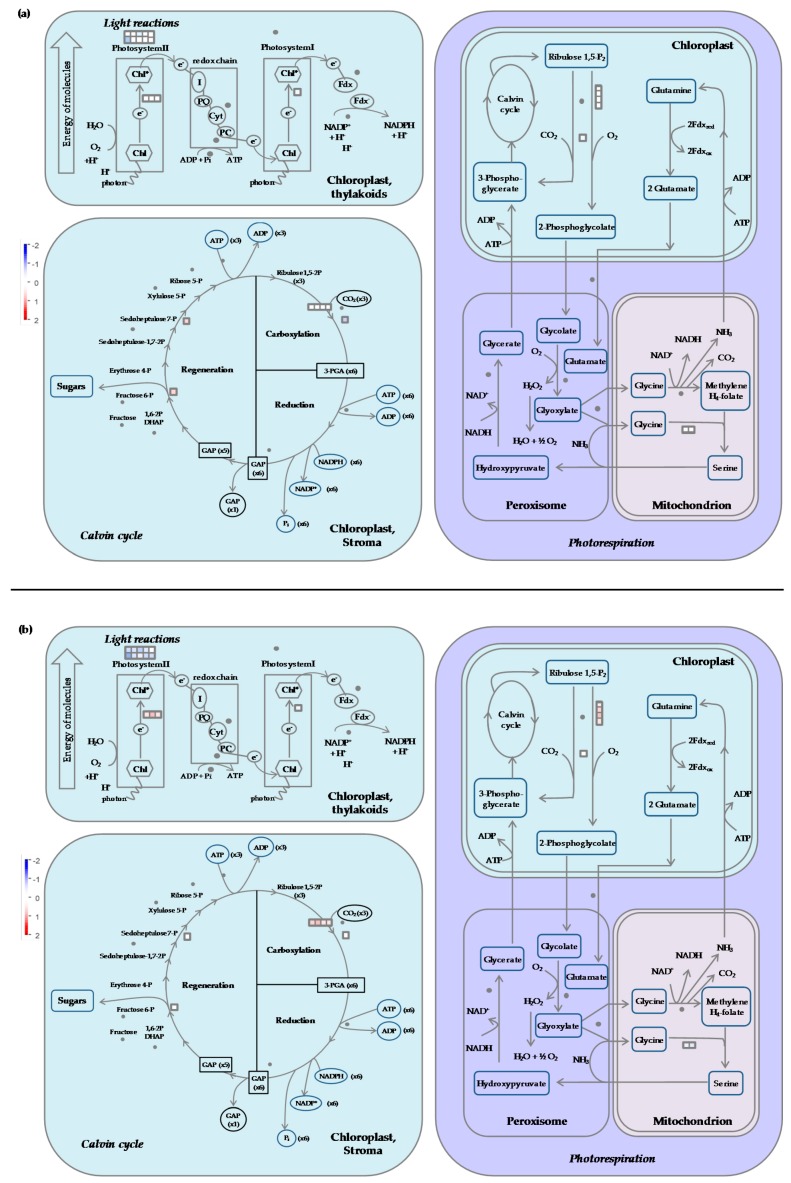
Visualization of the photosynthesis pathway modulation at (**a**) 8 h and (**b**) 2 h PS.

**Figure 7 ijms-20-01130-f007:**
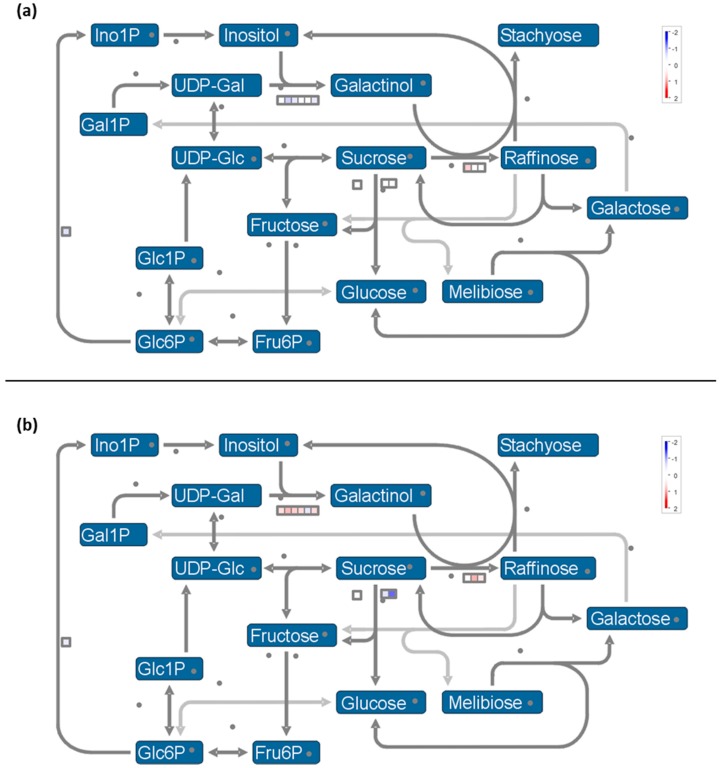
Mapman visualization of raffinose metabolism pathway modulation at (**a**) 8 h and (**b**) 2 h PS.

**Figure 8 ijms-20-01130-f008:**
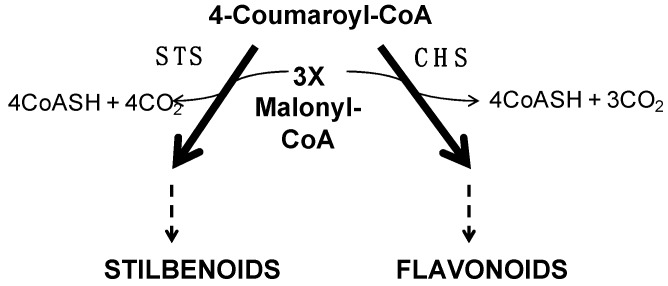
Pathways for stilbenoid and flavonoid biosyntheses adapted from Kodan et al. [159]. Stilbene synthase (STS) and chalcone synthase (CHS), respectively, led to stilbenoid and flavonoid biosynthesis from 4-Coumaroyl-CoA with three malonyl-CoAs.

**Figure 9 ijms-20-01130-f009:**
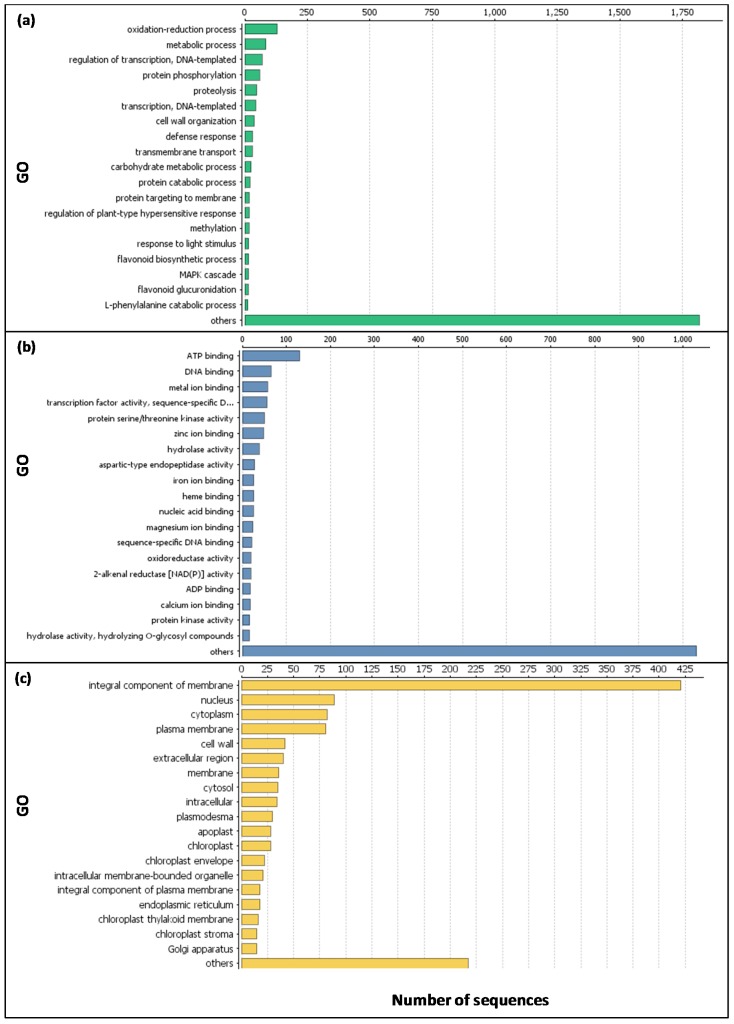
Global data analyses using Blast2GO software (BioBam Bioinformatics, Valencia, Spain). The X-axis indicates the number of transcripts in a sub-category and the Y-axis indicates the sub-category. (**a**) Biological process; (**b**) molecular function and (**c**) cellular component GO terms.

**Figure 10 ijms-20-01130-f010:**
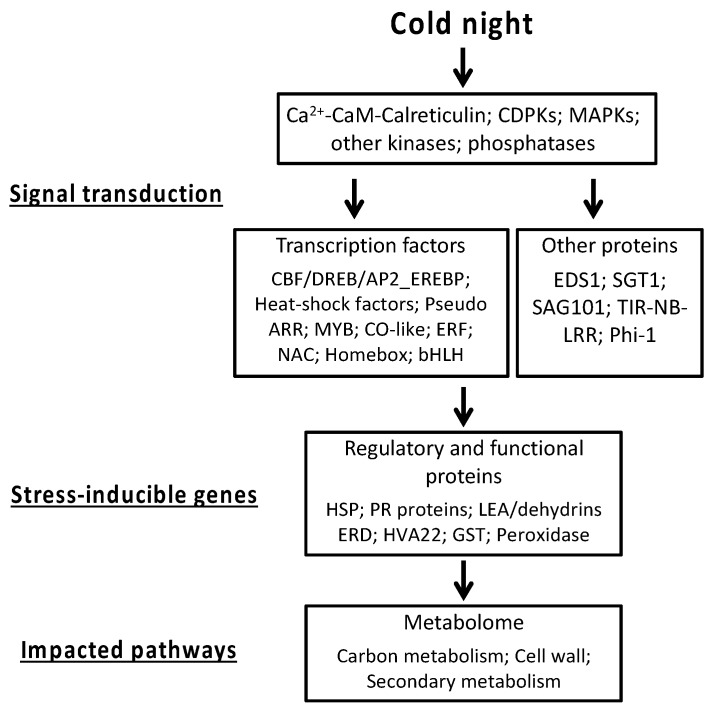
Schematic representation of the major mechanisms involved in the response of the *Vitis* flower to a cold night. Signal transduction pathway, stress-inducible genes, and major impacted pathways following the exposure of grapevine flowers to a cold night.

**Table 1 ijms-20-01130-t001:** Major functional categories highly impacted by the cold night at 8 h according to Corto software.

8 h		
BIN	Up-Regulated Process	Adjusted *p*-Value
10	cell wall	1.7110 × 10^−4^
10.7	cell wall.modification	5.3204 × 10^−4^
16.8	secondary metabolism.flavonoids	6.3979 × 10^−3^
16.8.3	secondary metabolism.flavonoids.dihydroflavonols	7.0367 × 10^−3^
16.8.3.1	secondary metabolism.flavonoids.dihydroflavonols, .dihydrokaempferol 4-reductase	5.3735 × 10^−3^
20	stress	2.2949 × 10^−8^
20.1	stress.biotic	3.8506 × 10^−3^
20.1.7	stress.biotic.PR-proteins	6.15 × 10^−3^
20.2	stress.abiotic	8.0636 × 10^−6^
20.2.1	stress.abiotic.heat	3.0648 × 10^−7^
24	Biodegradation of Xenobiotics	2.1821 × 10^−3^
27	RNA	4.0490 × 10^−4^
27.1.1	RNA.processing.splicing	8.2208 × 10^−3^
27.3	RNA regulation of transcription	1.5298 × 10^−4^
27.3.3	RNA regulation of transcription.AP2_EREBP, APETALA2_Ethylene-responsive element binding protein family	9.5059 × 10^−13^
27.3.66	RNA regulation of transcription.Psudo ARR transcription factor family	4.7432 × 10^−7^
29	protein	3.4384 × 10^−3^
29.5.1	protein degradation.subtilases	9.6717 × 10^−6^
30.1	signaling in sugar and nutrient physiology	5.8130 × 10^−5^
30.3	signaling.calcium	1.3157 × 10^−3^
34.3	transport.amino acids	5.32 × 10^−4^
**BIN**	**Down-Regulated Process**	**Adjusted *p*-Value**
3	minor CHO metabolism	2.8967 × 10^−3^
3.1	minor CHO metabolism.raffinose family	2.9243 × 10^−5^
3.1.1	minor CHO metabolism.raffinose family.galactinol synthases	2.4621 × 10^−6^
3.1.1.2	minor CHO metabolism.raffinose family.galactinol synthases.putative	2.4621 × 10^−6^
5.3	Fermentation.ADH	5.7860 × 10^−3^
8.3	TCA_org transformation.carbonic anhydrases	1.2604 × 10^−4^
10	cell wall	2.2613 × 10^−4^
10.6	cell walldegradation	5.4979 × 10^−4^
10.6.3	cell wall.degradation.pectate lyases and polygalacturonases	1.3581 × 10^−5^
10.7	cell wall.modification	3.0702× 10^−3^
16	secondary metabolism	3.2141 × 10^−12^
16.1	secondary metabolism.isoprenoids	4.1006 × 10^−13^
16.1.4	secondary metabolism.isoprenoids.carotenoids	4.4294 × 10^−3^
16.1.5	secondary metabolism.isoprenoids.terpenoids	3.7027 × 10^−16^
16.8	secondary metabolism.flavonoids	1.7181 × 10^−4^
16.8.1	secondary metabolism.flavonoids.anthocyanins	2.9899 × 10^−5^
16.8.1.12	secondary metabolism.flavonoids, anthocyanins.anthocyanidin 3-*O*-glucosyltransferase	2.7661 × 10^−4^
16.8.1.21	secondary metabolism.flavonoids, anthocyanins.anthocyanin 5-aromatic acyltransferase	1.4022 × 10^−3^
17.8	hormone metabolism.salicylic acid	2.0278 × 10^−3^
17.8.1	hormone metabolism.salicylic acid.synthesis-degradation	1.8638 × 10^−3^
19	tetrapyrrole synthesis	2.5726 × 10^−3^
27.3.26	RNA regulation of transcription.MYB-related transcription factor family	6.6359 × 10^−6^
27.3.7	RNA regulation of transcription.C2C2(Zn) CO-like, Constans-like zinc finger family	2.0362 × 10^−8^
30.11	signaling light	3.7892 × 10^−3^
34	transport	6.5478 × 10^−5^
34.19	transport.major intrinsic proteins	2.0278 × 10^−3^
34.4	transport.nitrate	5.7860 × 10^−3^
34.6	transport.sulphate	1.9185 × 10^−4^

**Table 2 ijms-20-01130-t002:** Major functional categories highly impacted by the cold night at 2 h PS according to the Corto software.

2 h PS		
BIN	Up-Regulated Process	Adjusted *p*-Value
1.3.1	PS.calvin cycle.rubisco large subunit	2.4217 × 10^−5^
3	minor CHO metabolism	2.4623 × 10^−3^
3.1	minor CHO metabolism.raffinose family	6.9210 × 10^−7^
3.1.1	minor CHO metabolism.raffinose family.galactinol synthases	7.6070 × 10^−6^
3.1.1.2	minor CHO metabolism.raffinose family.galactinol synthases.putative	7.6070 × 10^−6^
11.9.3.5	lipid metabolism.lipid degradation.lysophospholipases.phosphoinositide phospholipase C	4.3982 × 10^−3^
16.2	secondary metabolism.phenylpropanoids	1.3950 × 10^−3^
16.2.1	secondary metabolism.phenylpropanoids.lignin biosynthesis	1.4851 × 10^−4^
16.2.1.10	secondary metabolism.phenylpropanoids.lignin biosynthesis.CAD	5.2999 × 10^−6^
16.8.3	secondary metabolism.flavonoids.dihydroflavonols	4.3804 × 10^−3^
17	hormone metabolism	1.2461 × 10^−3^
17.1.3	hormone metabolism.abscisic acid.induced-regulated-responsive-activated	1.5786 × 10^−3^
17.8	hormone metabolism.salicylic acid	8.5038 × 10^−4^
17.8.1	hormone metabolism.salicylic acid.synthesis-degradation	7.4185 × 10^−4^
20	stress	3.7621 × 10^−7^
20.1	stress.biotic	1.2488 × 10^−8^
20.1.2	stress.biotic.receptors	1.8208 × 10^−3^
20.1.3	stress.biotic.signaling	2.9375 × 10^−8^
20.1.7	stress.biotic.PR-proteins	4.1661 × 10^−3^
20.2.3	stress.abiotic.drought_salt	4.8258 × 10^−4^
26.12	misc.peroxidases	2.0486 × 10^−3^
26.9	misc.glutathione S transferases	1.5046 × 10^−3^
27.3	RNA.regulation of transcription	7.5119 × 10^−4^
27.3.22	RNA.regulation of transcription.HB,Homeobox transcription factor family	5.9729 × 10^−5^
27.3.27	RNA.regulation of transcription.NAC domain transcription factor family	1.6861 × 10^−4^
27.3.6	RNA.regulation of transcription.bHLH,Basic Helix-Loop-Helix family	7.6192 × 10^−4^
28.1.3	DNA.synthesis_chromatin structure.histone	1.2067 × 10^−5^
29.2	protein.synthesis	2.6618 × 10^−3^
29.5.11	protein.degradation.ubiquitin	2.3331 × 10^−12^
29.5.9	protein.degradation.AAA type	1.3294 × 10^−3^
30	signaling	2.6317 × 10^−4^
30.2	signaling.receptor kinases	6.4535 × 10^−4^
30.2.20	signaling.receptor kinases.wheat LRK10 like	1.7662 × 10^−4^
30.2.99	signaling.receptor kinases.misc	9.1461 × 10^−4^
30.3	signaling.calcium	5.1743 × 10^−4^
34.13	transport.peptides and oligopeptides	1.2525 × 10^−4^
34.19	transport.Major Intrinsic Proteins	9.0675 × 10^−5^
34.19.1	transport.Major Intrinsic Proteins.PIP	5.5615 × 10^−4^
**BIN**	**Down-Regulated Process**	**Adjusted *p*-Value**
1	PS	4.5585 × 10^−5^
1.1	PS.lightreaction	2.8822 × 10^−5^
1.1.1	PS.lightreaction.photosystem II	3.0443 × 10^−8^
1.1.1.1	PS.lightreaction.photosystem II.LHC-II	4.7652 × 10^−15^
2.1.2	major CHO metabolism.synthesis.starch	4.1047 × 10^−3^
2.2.1.3.3	major CHO metabolism.degradation.sucrose.invertases.vacuolar	2.5554 × 10^−4^
10	cell wall	1.7409 × 10^−20^
10.2	cell wall.cellulose synthesis	8.9264 × 10^−10^
10.2.1	cell wall.cellulose synthesis.cellulose synthase	1.7420 × 10^−6^
10.6	cell wall.degradation	4.2186 × 10^−10^
10.6.3	cell wall.degradation.pectate lyases and polygalacturonases	1.6981 × 10^−7^
10.7	cell wall.modification	8.4556 × 10^−9^
11.1	lipid metabolism.FA synthesis and FA elongation	2.8132 × 10^−3^
13	amino acid metabolism	1.3271 × 10^−5^
13.1	amino acid metabolism.synthesis	1.1606 × 10^−5^
13.1.2	amino acid metabolism.synthesis.glutamate family	1.2995 × 10^−3^
13.1.5.1	amino acid metabolism.synthesis.serine-glycine-cysteine group.serine	3.6731 × 10^−3^
13.1.5.1.1	amino acid metabolism.synthesis.serine-glycine-cysteine group.serine.phosphoglycerate dehydrogenase	1.5008 × 10^−3^
16	secondary metabolism	2.0116 × 10^−19^
16.1	secondary metabolism.isoprenoids	1.0312 × 10^−7^
16.1.5	secondary metabolism.isoprenoids.terpenoids	3.5231 × 10^−11^
16.2	secondary metabolism.phenylpropanoids	7.0813 × 10^−12^
16.2.1	secondary metabolism.phenylpropanoids.lignin biosynthesis	1.2317 × 10^−12^
16.2.1.1	secondary metabolism.phenylpropanoids.lignin biosynthesis.PAL	3.0248 × 10^−20^
16.8	secondary metabolism.flavonoids	2.2763 × 10^−6^
16.8.2	secondary metabolism.flavonoids.chalcones	5.3110 × 10^−5^
16.8.2.1	secondary metabolism.flavonoids.chalcones.naringenin-chalcone synthase	1.3583 × 10^−4^
16.8.2.3.1	secondary metabolism.flavonoids.chalcones.pterostilbene synthase	4.0714 × 10^−6^
16.8.3	secondary metabolism.flavonoids.dihydroflavonols	4.2622 × 10^−3^
20.1	stress.biotic	4.9962 × 10^−5^
20.2	stress.abiotic	1.8297 × 10^−5^
20.2.1	stress.abiotic.heat	1.7390 × 10^−7^
29	protein	1.5050 × 10^−3^
29.5.1	protein.degradation.subtilases	3.8388 × 10^−3^
29.5.11	protein.degradation.ubiquitin	1.8703 × 10^−3^
31	cell	2.6211 × 10^−3^
33.99	development.unspecified	1.7360 × 10^−3^
34	transport	4.7864 × 10^−12^
34.10	transport.nucleotides	1.0332 × 10^−3^
34.4	transport.nitrate	1.9522 × 10^−3^
34.6	transport.sulphate	1.0111 × 10^−4^
34.99	transport.misc	3.4894 × 10^−6^

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
