# Peer review of "On a Cold Night: Transcriptomics of Grapevine Flower Unveils Signal Transduction and Impacted Metabolism"

_ijms, 2019, doi:10.3390/ijms20051130_

Round 1
Reviewer 1 Report
Dear authors,
I really appreciated the efforts made for improving the manuscript and I think that your work is now ready for publication.
Author Response
We would like to thank the reviewer for his compliments and the approval of our work.
Reviewer 2 Report
In this manuscript author did transcript analysis of grapevine flower that unveils signal transduction and impacted metabolism.
I have few comments to make.
1. Please functionally validate at least one gene for stress tolerance.
2. Put validation of expression analysis in the main figure instead supplementary figure.
Author Response
1.While this is certainly an interesting and valid suggestion, we would here like to highlight that our manuscript is centered on transcriptional dynamics measured via the microarray technology. Our purpose is to shed light on the gene expression changes by adopting a broad angle. It would not be possible to perform a functional validation in the 10 days-time window allowed before resubmission and, also, our study will pave the way to such follow-up studies, as described in the Abstract. It should also be noted that the validation via heterologous expression approaches in other plant species/organisms has already been undertaken for some of the genes we have described, notably LEA proteins (Yu et al. 2016, Biochem Biophys Res Commun. 478:703-709. doi: 10.1016/j.bbrc.2016.08.010), dehydrins (Zhou et al. 2017, AMB Express. 7:182. doi:10.1186/s13568-017-0483-1), GST (Takesawa et al. 2002, Mol. Breed 9:93-101. doi: 10.1023/A:1026718308155).
2.done.
Moreover, we conducted a careful reading to remove the last mistakes.
Round 2
Reviewer 2 Report
I am happy with the authors reply and this MS can be accepted in its current form.
This manuscript is a resubmission of an earlier submission. The following is a list of the peer review reports and author responses from that submission.
Round 1
Reviewer 1 Report
In this manuscript, the authors described the transcriptomic of grapevine flower in cold conditions that unveil signal transduction and impacted metabolism. Manuscript lack many grammatic errors and average writing. For the betterment of this MS I have comments to make:
Major:
1. All manuscript is only about Mapman analysis which is just online web tools. MS lacks wet lab experiments. I would be nice if authors try to validate some really DEGs by real-time PCR and made some kind of signaling network. I am not sure this kind of work can be accepted in a reputed journal like IJMS.
2. Introduction part is very-very short. Explain previous studies in detail and add recent examples such as:
a. Genome-wide analysis of gene expression profiling revealed that COP9 signalosome is essential for correct expression of Fe homeostasis genes in Arabidopsis.
b. Genome-Wide Identification and Analysis of Genes, Conserved between japonica and indica Rice Cultivars, that Respond to Low-Temperature Stress at the Vegetative Growth Stage. Front. Plant Sci. 8:1120. doi: 10.3389/fpls.2017.01120
c. Genome-Wide Analysis of the PYL Gene Family and Identification of PYL Genes That Respond to Abiotic Stress in Brassica napus.
Minor:
At Line3 change unveals to unveils.
L13 Environnemental to Environmental.
L42 longer term to longer-term.
L51 unfavourable to unfavorable.
L72 What is the meaning of this line? LEA and HSP have been showare known to be involved in protecting macromolecules, such as enzymes and lipids.
L81 low temperature to low-temperature.
L122 one time to one-time.
L139 donwregulated to downregulated.
L312 Senescence Associated to Senescence-Associated.
L400 imbalance leads to an to imbalance leads to a.
L473 transglucosylases or transglycosylases?
L613 high density to high-density.
L621 Two micron to Two-micron.
L667 gene specific to gene-specific.
L669 budbreak or bud break?
Author Response
Point 1. All manuscript is only about Mapman analysis which is just online web tools. MS lacks wet lab experiments. I would be nice if authors try to validate some really DEGs by real-time PCR and made some kind of signaling network. I am not sure this kind of work can be accepted in a reputed journal like IJMS.
Response 1. We have actually performed a validation using qPCR on a set of 16 genes. The results show a good correlation between the microarray analysis and the qPCR results. Please see Additional File 1 and Table 1 for the list of primers used. As for the signalling network, a STRING network could be envisaged. However, in this case, it should be noted that in order to find some interesting and not yet described candidates, the threshold should be lowered, which makes the result just indicative, since functional studies would be required. We have instead added further insights into the cell wall part, where a GOE analysis using ClueGO (within Cytoscape) has been performed to highlight the up-regulation of sporopollenin biosynthesis. This is interesting since this suggests a response to cold involving the synthesis of the biopolymer sporopollenin.
Point 2. Introduction part is very-very short. Explain previous studies in detail and add recent examples such as:
Response 2. Considering the high number of cited references (167) we apologize for not citing the above references since they are not dealing with inflorescence organs and/or woody plants.
Minor:
At Line3 change unveals to unveils.
Done
L13 Environnemental to Environmental.
Done
L42 longer term to longer-term.
Done
L51 unfavourable to unfavorable.
Done
L72 What is the meaning of this line? LEA and HSP have been showare known to be involved in protecting macromolecules, such as enzymes and lipids.
We apologize for the mistake. We have corrected it.
L81 low temperature to low-temperature.
Done
L122 one time to one-time.
We have corrected time point to time-point.
L139 donwregulated to downregulated.
Done
L312 Senescence Associated to Senescence-Associated.
Done
L400 imbalance leads to an to imbalance leads to a.
Actually the noun starts with a vowel and it should be an.
L473 transglucosylases or transglycosylases?
Done
L613 high density to high-density.
Done
L621 Two micron to Two-micron.
Done
L667 gene specific to gene-specific.
Done
L669 budbreak or bud break?
We have modified to bud break
Reviewer 2 Report
General comment
In this work, Sawicki and co-workers discuss changes due to transcriptional reprogramming events taking place in grapevine flowers exposed to low temperatures and subsequent recovery. Although the manuscript is well-written and focuses on a phenomenon still poorly explored in grapevine, this study risks to remain limited to a description of transcriptomic changes occurring upon flower exposure to a cold night. This is the main reason why I retain that additional information is needed to strengthen the results provided and improve the overall quality of the manuscript prior to publication. More details are following indicated in the evaluation report.
- Several aspects related to alteration of specific metabolic pathways, especially the signal transduction processes here associated to cold stress response for the first time, should be supported better and discussed more in depth as well. To this aim, I suggest to include in the main text only a selection of the many MapMan charts displaying changes in plant metabolisms, and I would rather report pictures showing expression profiles of specific candidate genes among those analyzed by Real Time PCR in the tested experimental condition. This is important to focus more clearly on those transcriptional differences playing a key role in the signalling cascades discussed throughout the article. An example can be the validated genes encoding the main enzymes of stilbene and terpene biosynthesis and/or genes involved in cell wall modifications, sugar metabolism (ie. starch synthase, beta-amylase and invertase encoding genes) and transport (ie. sucrose transporter genes).
- Another issue that I warmly suggest to address is the quantification of at least some metabolites (examples can be single sugars, starch and/or secondary metabolites, such as stilbenoids and terpenes) putatively exerting a key signalling function in the regulation of molecular mechanisms underlying cold stress response in grapevine flowers. Accordingly, I would spend words in describing these metabolic changes in support to the expression differences of the related genes.
I think that this additional set of information would greatly help to provide more insights unravelling the functional links between the observed transcriptional changes and plant physiological response to cold temperature, thus making the picture of hypothesized mechanisms more compelling, especially for what it concerns those specifically associated to cold stress for the first time.
- Finally, I would shorten the Results and Discussion section a bit and, more importantly, since the authors merged results and discussion, I recommend to find more appealing headings for the diverse paragraphs, at least for the main ones. This would help to provide an idea of the main achievements (especially in the case of the new molecular responses highlighted by the study; ie. transcriptional changes in terpene metabolism) immediately by reading the main titles reported throughout the section.
Other points:
- The authors well outlined the novelty of their work in the Introduction section, underlining that this is the first study on transcriptomic effects occurring in grapevine flowers following exposure to cold stress. However, how about similar studies performed in other crop species, specifically in the woody ones? In case this work was the first reporting transcriptional changes due to cold stress at the flower level, not only in grapevine but overall in plants, this should be highlighted in the Introduction of the manuscript in order to stress the originality of the study or, on the contrary, to provide more information on the current state of the art.
- Figures should be improved in terms of resolution and/or by magnifying some details, such as the really small blue/red squares representing the differentially expressed genes in the MapMan charts, especially in the case of figures 6 and 7.
- Table 2 and 3: I suggest to find a different and more straightforward way to present these data than tables; a good alternative may be pie charts showing the distribution of up- and down- regulated functional gene categories.
- Captions to supplementary tables should be provided at least in the excel files.
- Table 1 listing oligonucleotides for RT-qPCR analysis should be shifted in the supplementary files and CRIBI GGDB 12X V1 accessions should be provided rather than NCBI accessions, especially considering that both in the main text and in supplementary tables the reported transcript accessions refer to the first database.
- lines 592-600: how many plants were utilized for the experiment? how many of them were subjected to cold stress and/or recovery? How many control plants were employed throughout the experiments?
- lines 603-604: was the pool of 6 inflorescences obtained from a single plant in turn representing one biological replicate or were the inflorescences sampled from different plants and then pooled together?
- line 73: What did the authors mean for 'showare'?
- line 598: please, substitute ‘during’ with ‘for’.
Author Response
General comment
Point 1 Response. This study is effectively a descriptive work on transcriptomic changes. In order to improve significantly the quality of our manuscript, we have taken into account most of your suggestions.
Point 2. Response. In the updated manuscript we considered the suggestions proposed by the referee. Thus, we reformulated/deleted different parts of the manuscript.
Point 3. Response. Unfortunately, we do not have the necessary material for this complementary analysis. We cannot therefore respond favourably to this request.
Point 4. Response. As recommended, we tried to find comprehensive and appropriate headings to highlight the main information of paragraphs for smooth reading. We hope that this new version will fit with the reviewer requests.
Other points:
Point 5. Response. Thanks for the compliment. We reformulated our introduction to highlight the originality and the importance of this work. Even if Rowland et al. (2012) studied the transcriptomic sequences on flower buds in response to cold, this is the first real global transcriptomic analysis dealing with the effect of cold stress in the inflorescence of woody plant, at the crucial developmental stage for future fruit development (female meiosis). This analysis is consequently crucial for understanding the mechanisms of flower abortion due to cold stress in perennial crops.
Point 6. Response. We improved the figures as requested.
Point 7. Response. Considering the number of impacted categories, the best way to present the results is still table. Moreover, table permits also to present the BIN, and the adjusted p-value.
- Captions to supplementary tables should be provided at least in the excel files.
done
- Table 1 listing oligonucleotides for RT-qPCR analysis should be shifted in the supplementary files and CRIBI GGDB 12X V1 accessions should be provided rather than NCBI accessions, especially considering that both in the main text and in supplementary tables the reported transcript accessions refer to the first database.
done
Point 8. lines 592-600: how many plants were utilized for the experiment? how many of them were subjected to cold stress and/or recovery? How many control plants were employed throughout the experiments?
Response 8. Six inflorescences from 6 different plants were used for each treatment and each time point. We added the precision in the text.
- lines 603-604: was the pool of 6 inflorescences obtained from a single plant in turn representing one biological replicate or were the inflorescences sampled from different plants and then pooled together?
done
Point 9. line 73: What did the authors mean for 'showare'?
Response 9. We apologize for the mistake. We have corrected it.
- line 598: please, substitute ‘during’ with ‘for’.
done
Reviewer 3 report
The main objectives of the study were to identify changes in transcripts levels after a cold night at the female meiosis time, and during recovery 2h following the end of the cold night using cuttings from grapevine canes.
Before delving into specifics of the study, the first question that arises has to do with only two time points as noted above. The overarching paradigm is that recovery from a cold night (stress) is going to be a continuous process upon exposure to 25 C in the light that does not begin at 2 h post stress and similarly does not reach an endpoint at that sampling. Is there clear empirical evidence to show that 2 h post stress is when it is known that megagametogenesis is either showing deleterious effects of cold or signs of recovery from stress? Why choose 2 h is the basic question, does it represent a critical moment in either recovery or death? Thus the data in this study is but a single snapshot of a complex dynamic cellular process. Ideally, a well designed time course with multiple time points would be perhaps more informative.
There are four sampling points (1) control at the end of 8 h 19 C night and (2) 2 h post 19 C at 25 C in the light, and for the treatment (3) at the end of 8 h 0 C night and (4) 2 h post stress at 25 C. A biological sample consists of 6 inflorescences, and there are two biological replicates at each sampling for the control and the cold treated inflorescences. The authors' state: "The differential analysis is based on the log-ratios averaging over the duplicate probes and over the technical replicates. Hence the numbers of available data for each gene equals the number of biological replicates and are used to apply a moderated t-test [165]." Therefore, data for each of the four samplings has an N=2. After all of the data processing, what is the evidence that all the variances for the means where N=2 are really all the same or even somewhat similar? It seems to be remarkable that the signal values for more than 29,000 genes and two different temperature conditions would have similar or equal variances over perhaps a 103 or 105 signal range. Furthermore, with a true sample of N=2, how does one arrive at a conclusion that the recorded mean signal values of N=2 are without exception normally distributed? What is the potential consequence of "smoothing the variances by computing empirical Bayes posterior means?" Perhaps deriving the ratio of treatment/control solves or permits some of assumptions. These are questions I cannot answer with certainty as I have little knowledge of statistics. However, the overarching concern is this study is based on 4 samples each with 2 biological replicates that were created from 6 inflorescences each, and therefore is based on a total 24 inflorescences. It is possible that more biological replicates would yield a more robust and reliable dataset to conduct the statistical analyses with perhaps different realized outcomes.
The objective of the study was to reveal changes in gene expression of grapevine flowers in response to a low temperature night compared to a control condition of 19 C. Information from such an experimental design would be novel and provide new understanding of how low temperatures affect megagametogenesis.
In tables 2 and 3, the p-values are adjusted or unadjusted values for multiple comparisons?
What is the significance of altered expression of genes encoding proteins and enzymes of photosynthesis? Are the inflorescence tissues green and photosynthetic?
Perhaps the saving grace for this study given its minimal replication and sampling regime is the fact that many genes known to be responsive to low temperature exposure in plants are also found to be responsive in the inflorescence tissues of grape.
My overall view of the work described in this manuscript is that it serves the purpose of a pilot study where a small experimental design with limited sampling is conducted and reveals responses and outcomes that justify a much larger and more comprehensive study to confirm some the potentially novel results presented and to provide a more complete understanding of the complex process of grapevine inflorescence tissues in response to low temperature exposure.
Author response
The main objectives of the study were to identify changes in transcripts levels after a cold night at the female meiosis time, and during recovery 2h following the end of the cold night using cuttings from grapevine canes.
Point 1. Before delving into specifics of the study, the first question that arises has to do with only two time points as noted above. The overarching paradigm is that recovery from a cold night (stress) is going to be a continuous process upon exposure to 25 C in the light that does not begin at 2 h post stress and similarly does not reach an endpoint at that sampling. Is there clear empirical evidence to show that 2 h post stress is when it is known that megagametogenesis is either showing deleterious effects of cold or signs of recovery from stress? Why choose 2 h is the basic question, does it represent a critical moment in either recovery or death? Thus the data in this study is but a single snapshot of a complex dynamic cellular process. Ideally, a well designed time course with multiple time points would be perhaps more informative.
Response 1. The transcriptome recovering is quite fast and the important transcripts for adaptation/recovery are still induced more time (Lyon et al. 2016; Crisp et al. 2017). Following a stress, “common” modified genes, which create a background noise at transcriptomic level, regain their expression baseline following 30-60 minutes after the stress. In this respect, we have chosen to follow the gene expression two hours after the end of the cold night. Thus, we could be sure that modified genes are really important and involved in the stress response/stress recovery.
Point 2. There are four sampling points (1) control at the end of 8 h 19 C night and (2) 2 h post 19 C at 25 C in the light, and for the treatment (3) at the end of 8 h 0 C night and (4) 2 h post stress at 25 C. A biological sample consists of 6 inflorescences, and there are two biological replicates at each sampling for the control and the cold treated inflorescences. The authors' state: "The differential analysis is based on the log-ratios averaging over the duplicate probes and over the technical replicates. Hence the numbers of available data for each gene equals the number of biological replicates and are used to apply a moderated t-test [165]." Therefore, data for each of the four samplings has an N=2. After all of the data processing, what is the evidence that all the variances for the means where N=2 are really all the same or even somewhat similar?
Response 2. Performing tests on microarray data is very tricky and was an open question for the statisticians. It is true that the number of observations per gene is small but the number of genes being large, methods of differential analysis were developed for this specific context. According to the paper of Jeanmougin et al 2010 (PlosOne), we used the moderated t-test to perform the differential analysis. They developed the empirical Bayes method to model the variance, where extra information is borrowed from the ensemble of genes for inference about each individual gene. This is possible because the number of genes is large. When we performed the analysis with this method on this dataset, the method concluded that there is no evidence of the specific variance between probes. These conclusions are the outputs of the method not ours.
Point 3. It seems to be remarkable that the signal values for more than 29,000 genes and two different temperature conditions would have similar or equal variances over perhaps a 103 or 105 signal range. Furthermore, with a true sample of N=2, how does one arrive at a conclusion that the recorded mean signal values of N=2 are without exception normally distributed? What is the potential consequence of "smoothing the variances by computing empirical Bayes posterior means?" Perhaps deriving the ratio of treatment/control solves or permits some of assumptions. These are questions I cannot answer with certainty as I have little knowledge of statistics.
Response 3. The number of replicates is important for the power of the statistical tests. For sure, with more replicates, results would be more precise and we most certainly would have detected more differentially expressed genes. However, the experimental design for this project is standard according to what we observe in other projects, and does not question the obtained results: the genes that were deemed differentially expressed in our experimental designed indeed are differentially expressed.
Point 4. However, the overarching concern is this study is based on 4 samples each with 2 biological replicates that were created from 6 inflorescences each, and therefore is based on a total 24 inflorescences. It is possible that more biological replicates would yield a more robust and reliable dataset to conduct the statistical analyses with perhaps different realized outcomes.
Response 4. For each time point 24 inflorescences, representing 24 plants at the same developmental stage (female meiosis occurring approximately 20 hours in the plant’s life), were used. We think that it consists a robust dataset.
Point 5. The objective of the study was to reveal changes in gene expression of grapevine flowers in response to a low temperature night compared to a control condition of 19 C. Information from such an experimental design would be novel and provide new understanding of how low temperatures affect megagametogenesis.
In tables 2 and 3, the p-values are adjusted or unadjusted values for multiple comparisons?
Response 5. We present adjusted p-values. This has been modified in the table.
Point 6. What is the significance of altered expression of genes encoding proteins and enzymes of photosynthesis? Are the inflorescence tissues green and photosynthetic?
Response 6. The inflorescence are photosynthetic organs (Sawicki et al. 2015, 2017), it’s the reason why we focused on genes involved in photosynthesis process.
Point 7. Perhaps the saving grace for this study given its minimal replication and sampling regime is the fact that many genes known to be responsive to low temperature exposure in plants are also found to be responsive in the inflorescence tissues of grape.
Response 7. Your observation supports that dataset of 24 plants represents a robust sample and that our experimental design has been properly conducted.
Point 8. My overall view of the work described in this manuscript is that it serves the purpose of a pilot study where a small experimental design with limited sampling is conducted and reveals responses and outcomes that justify a much larger and more comprehensive study to confirm some the potentially novel results presented and to provide a more complete understanding of the complex process of grapevine inflorescence tissues in response to low temperature exposure.
Response 8. With this global approach, we identified a maximum of cold-induced changes in grapevine flowers at transcriptomic level. We hope that these data will provide the scientific community with a solid basis for future investigations regarding the impact of stress on flower and related abortion mechanisms.
Round 2
Reviewer 1 Report
I am still not happy with the response of the authors. Still, the manuscript is all about mapman analysis and does not have enough molecular as well as
physiological data. I am afraid in its current format I am not going to accept this in IJMS.
Author Response
We feel there is nothing we can do to persuade the Reviewer about the validity of our results. We do however wish to highlight the complexity of the study and the technical challenge in preparing the samples. We are persuaded about the novelty of our results (no comparable papers are present, to the best of our knowledge, in the literature). Also, the other Reviewer highlighted the innovative aspect of this study.
Reviewer 2 Report
I appreciated the efforts made by the authors to improve the manuscript, and I noticed that the quality of figures was improved as well. However, my previous suggestions have only partially been addressed and I retain that the manuscript requires other revisions prior to publication.
1) I can understand that, in a study set up for transcriptomic analysis exclusively, the authors did not plan to quantify metabolites, thus they do not have enough material to run the additional analyses that I suggested. Anyway, I must say that I agree with the authors’ explanation only because very few information is available on flower responses to cold stress in grapevine, and in plant overall. Secondly, I think that this study provides interesting insights paving the way for future investigations on this topic. This is the only reason why I can accept the lack of more functional information providing a deeper characterization of the transcriptional changes observed here.
2) The authors did not address my suggestion to report in the main text some results of specific group of genes profiled by Real Time PCR. Since interesting overviews of down and up regulated genes were unveiled by bioinformatic elaboration of microarray data, I think that providing some graphs showing the expression profiles of key genes (analysed by RTqPCR) will help to highlight better the most important transcriptional differences discussed by the authors and, secondly, it will greatly help the reader to interpret the data in a straightforward way than a list of MapMan charts only. I warmly suggest to address this point, re-considering the detailed indications provided with my previous evaluation and related to this part.
3) As I have already suggested, I would shift Table 1 reporting oligonucleotide sequences in supplementary materials.
Other points:
I would suggest to add ‘key’ and ‘pathways’ in the title of the manuscript prior and after the words ‘signal transduction’;
lines 37-38: I suggest to write only ‘the up-regulation of genes encoding…’;
line 41: I would start this sentence with ‘Taken together, our results…’;
line 42: I would remove ‘the’ before ‘stress’ and I would add ‘grapevine’ before ‘flower’;
line 44: As some of the keywords are words already present in the title of the manuscript, I suggest to change them a bit. For instance: the authors may use ‘signaling cascades’ instead of ‘signal transduction’, candidate gene expression instead of transcriptomics (but only if the analysis of key genes by RTqPCR suggested at point 2) will be effectively reported in the main text), ‘primary’ can be removed as well;
lines 115-118: This part should be re-writed better, a suggestion maybe: ‘Moreover, with the exception of a previous study addressing cold-induced transcriptional changes in buds of blueberry, molecular response to cold stress are still poorly characterized at the flower level in plants. In particular, to our knowledge, the present study is the first to explore effects of cold in flowers of grapevine.’;
line 238: I would re-write this title as following proposed: ‘Genes encoding CBF transcription factors are induced by cold stress’;
line 263: I suggest to use ‘affects’ instead of ‘impacts’;
line 281: remove ‘also’ after ‘altered’;
line 323: I suggest to substitute the title with ‘Cold modulates the expression of genes involved in global stress responses’;
lines 390-391: I suggest to re-write this title as following proposed: ‘Transcripts involved in detoxification pathways are over-expressed by cold stress in grapevine flowers’;
lines 457-458: remove ‘Sugars’ and ‘period’ in the title;
lines 589-594: change ‘homolog’ to ‘homologue’;
line 619: remove ‘s’ at the end of ‘flavonoids’;
line 642: remove interestingly (this is an expression the authors can use to describe the results, but I would avoid its use in the paragraph title);
line 662: change ‘issued’ to ‘collected’;
Author Response
1) We would like to thank the reviewer for his understanding about our experiment plan and for the recognition of the work’s originality.
2) We added the requested graph (additional file 2) with the 15 selected genes expression profiles obtained by RT-qPCR. We hope that this adding will fit with the reviewer’s expectation. We have also implemented the text with these results.
3) We have transferred Table 1 into additional files.
Other points:
I would suggest to add ‘key’ and ‘pathways’ in the title of the manuscript prior and after the words ‘signal transduction’;
lines 37-38: I suggest to write only ‘the up-regulation of genes encoding…’;
done
line 41: I would start this sentence with ‘Taken together, our results…’;
done
line 42: I would remove ‘the’ before ‘stress’ and I would add ‘grapevine’ before ‘flower’;
done
line 44: As some of the keywords are words already present in the title of the manuscript, I suggest to change them a bit. For instance: the authors may use ‘signaling cascades’ instead of ‘signal transduction’, candidate gene expression instead of transcriptomics (but only if the analysis of key genes by RTqPCR suggested at point 2) will be effectively reported in the main text), ‘primary’ can be removed as well;
lines 115-118: This part should be re-writed better, a suggestion maybe: ‘Moreover, with the exception of a previous study addressing cold-induced transcriptional changes in buds of blueberry, molecular response to cold stress are still poorly characterized at the flower level in plants. In particular, to our knowledge, the present study is the first to explore effects of cold in flowers of grapevine.’;
Thank you for these suggestions, we have included all of them.
line 238: I would re-write this title as following proposed: ‘Genes encoding CBF transcription factors are induced by cold stress’;
done
line 263: I suggest to use ‘affects’ instead of ‘impacts’;
done
line 281: remove ‘also’ after ‘altered’;
done
line 323: I suggest to substitute the title with ‘Cold modulates the expression of genes involved in global stress responses’;
done
lines 390-391: I suggest to re-write this title as following proposed: ‘Transcripts involved in detoxification pathways are over-expressed by cold stress in grapevine flowers’;
done
lines 457-458: remove ‘Sugars’ and ‘period’ in the title;
done
lines 589-594: change ‘homolog’ to ‘homologue’;
done
line 619: remove ‘s’ at the end of ‘flavonoids’;
done
line 642: remove interestingly (this is an expression the authors can use to describe the results, but I would avoid its use in the paragraph title);
done
line 662: change ‘issued’ to ‘collected’;
done